# How Much Can RAG Help the Reasoning of LLM?

## Abstract

Retrieval-Augmented Generation (RAG) has gained significant popularity in modern Large Language Models (LLMs) due to its effectiveness in introducing new knowledge and reducing hallucinations. However, the deep understanding of RAG remains limited, how does RAG help the reasoning process and can RAG help improve the reasoning capability remains question. While external documents are typically considered as a method to incorporate domain-specific information, they also contain intermediate reasoning results related to the query, this suggests that documents could enhance the reasoning capability of LLMs, which has not been previously explored. In this paper, we investigate this issue in depth and find that while RAG can assist with reasoning, the help is limited. If we conceptualize the reasoning process as a tree with fixed depth, then RAG struggles to assist LLMs in performing deeper reasoning. Additionally, the information in the documents requires preprocessing to filter out noise. We demonstrate that this preprocessing is difficult to achieve simply fine-tuning of the LLM, it often necessitates numerous additional transformer layers to solve the problem. To simplify the problem, we propose DPrompt tuning, which effectively resolves the issue within just limited transformer layers, leading to improved performance.

## 1 Introduction

Large Language Models (LLMs) (Brown, 2020) have demonstrated remarkable capabilities across a variety of tasks, including text generation and question answering (Ouyang et al., 2022; Wei et al., 2022), code generation (Gu, 2023), and information retrieval (Dai et al., 2024). However, current LLMs often suffer from serious hallucinations (Huang et al., 2023) due to a lack of factual information. Moreover, the knowledge embedded within LLMs is encoded in their parameters (Yang et al., 2024), meaning that incorporating new knowledge requires further fine-tuning, which is both time-consuming and resource-intensive. Consequently, augmenting LLMs with external retrievers has led to significant performance improvements (Lewis et al., 2020; Zhao et al., 2024; Izacard et al., 2023).

Despite the widespread adoption of RAG in modern LLMs, a comprehensive understanding of how RAG aids inference remains an open question. Most researchers currently view RAG primarily as a method to provide domain-specific knowledge (Zhang et al., 2024a), often seeking to adapt LLMs to particular domains through RAG (Soudani et al., 2024). However, the impact of RAG on enhancing reasoning capacity has yet to be thoroughly investigated.

First, we aim to understand how RAG operates and whether it can improve the reasoning capabilities of LLMs. We can consider the LLM as computing $p(y|q)$, where $q$ represents the query and $y$ is the corresponding answer. In this context, retrieval-augmented generation can be expressed as $p(y|q, d_1, d_2, \ldots, d_k)$, where $d_i$ is the $i$-th document retrieved based on the query $q$. Additionally, the well-known prompting technique, Chain of Thought (CoT) (Wei et al., 2022), which significantly enhances the reasoning ability of LLMs, can be represented as $p(y|q, x_1, x_2, \ldots, x_k)$, where $x_i$ denotes the step-by-step reasoning results. Both CoT and RAG aim to incorporate additional information into the input to achieve better performance. It has been theoretically demonstrated and experimentally verified that CoT effectively improves the reasoning ability of LLMs (Feng et al., 2024; Wei et al., 2022; Chu et al., 2023). Thus, the question arises: **Can RAG also enhance the reasoning capabilities of LLMs?**

Fixed-depth transformer models with log precision can be simulated by constant-depth circuits (Merrill & Sabharwal, 2023), indicating that LLMs are limited to fixed-depth reasoning. When conceptualizing the reasoning path as a tree, the maximum depth remains constant (Merrill & Sabharwal, 2023). Chain of Thought (CoT) generates step-by-step reasoning or explanations rather than providing direct answers, formalized as $x_1 = f(x)$, $x_2 = f(x, x_1), \ldots, y = f(x, x_1, \ldots, x_k)$. This process allows CoT to effectively expand reasoning depth by execute $f$ multiple times, potentially reaching infinite depth with sufficient CoT steps (Feng et al., 2024; Li et al., 2024). In contrast, Retrieval-Augmented Generation (RAG) does not facilitate multiple inferences; instead, it retrieves existing relevant information to generate answers (Shi et al., 2023b). While RAG cannot stack LLM layers to enhance reasoning capability, the retrieved documents may include intermediate reasoning results, thereby reducing the number of layers needed for inference and enabling the LLM to tackle more complex problems, thus aiding its reasoning ability.

In this paper, we first demonstrate that RAG can enhance the reasoning ability of LLMs, enabling them to tackle more complex problems. If the model can initially solve problems with a reasoning depth of $l$, then with the addition of relevant document information, it could address problems with a reasoning depth of $l + c$, where $c$ is a constant determined by the quality of the retrieved information. This shows that **RAG can help to improve the reasoning capability of LLM.**

However, in real-world RAG scenarios, the information retrieved from documents is not always directly usable and often requires further processing because documents may contain noise information (Jiang et al., 2023b;a), and some documents may even be completely distracting, containing incorrect answers (Shi et al., 2023a; Wu et al., 2024). Such noise and distracting documents can negatively impact performance. While some works attempt to fine-tune the model to filter out noise (Xu et al., 2024) and distracting documents (Yan et al., 2024), these filtering processes may require additional reasoning depth and could ultimately harm overall performance. Also some works train another filter model (Yan et al., 2024; Asai et al., 2023), but this approach results in additional reasoning cost and makes it difficult to eliminate the inherent noise in the documents.

Therefore, a critical question arises: **How difficult is the filtering problem, and can we effectively solve it within a limited number of layers?** If the efforts to filter out noise is even more than the help RAG brings, RAG fail to improve the reasoning capability. First, we show that fine-tuning methods like LoRA also struggle to filter out noise without compromising reasoning capability, since filtering irrelevant tokens while maintaining the original attention patterns for relevant tokens proves challenging. This indicates that the filtering process cannot be effectively incorporated into the original reasoning steps of LLMs, necessitating additional reasoning depth for filtering, which can ultimately degrade the reasoning capability of the model.

Then we show that **judging the relevance of a token can hardly be implemented by limited number of transformer layers.** Considering the query *'Alice is exhausted but Bob is still excited; how does Bob feel?'* Here, *'exhausted'* is noise and should be excluded from the inference. However, assessing relevance necessitates considering the token *'Bob'* in the query alongside *'Alice'* the subject of *'exhausted'*. Thus, evaluating relevance requires the information from three or more tokens. Yet, the attention mechanism typically computes only pair-wise relationships, making it challenging to resolve this issue within a limited number of transformer layers (Sanford et al., 2024). With the number of transformer layers required for preprocessing increases, utilizing the document information for inference would help less, and even hurt the reasoning capability. To address this, we propose DPrompt tuning which transforms the triple-wise problem into a pair-wise one, and the pair-wise problem can be effectively tackled by limited layer of transformers, ultimately enhancing the reasoning ability of Retrieval-Augmented Language Models (RALM).

The main contributions of this paper are: (1) We demonstrate that RAG can enhance the reasoning ability of LLMs; however, this improvement is limited, allowing Retrieval-Augmented Language Models (RALMs) to solve problems that require slightly more reasoning depth than before; (2) We assert that fine-tuning the model to filter out noise while preserving its original functionality is challenging. This indicates that noise would lead to worse performance even after finetuning; (3) We further reveal that the inherent triple-wise nature of the filtering process may necessitate numerous transformer layers for effective document preprocessing, which could significantly degrade reasoning ability. To address this, we propose a method to transform the problem into a pair-wise format, making it easier to solve.

In Section 2, we explore the extent to which RAG can enhance the reasoning of LLMs, demonstrating that RAG has limited potential to improve reasoning capacity. In Section 3, we discuss the negative impact of noise on performance and the challenges associated with incorporating the filtering process into the reasoning process. In Section 4, we highlight the complexities involved in judging relevance, illustrating that solving this problem requires multiple layers, and we propose a method to simplify the process.

## 2 HOW DOES RAG HELP REASONING

In this section, we explore the extent to which RAG can enhance the reasoning ability of LLMs, aiming to deepen our understanding of its role in reasoning. As demonstrated by Merrill & Sabharwal (2023), transformer-based models are constrained to a fixed depth of reasoning due to the limited number of layers. Chain of Thought (CoT) facilitates step-by-step reasoning, effectively stacking multiple LLMs to address a single problem. Thus, the question is: **How can RAG assist during the reasoning process, and how much additional reasoning capacity can RAG provide to LLMs?**

### 2.1 THE FISSION OF RAG

If we consider the reasoning steps of an LLM as a tree, the maximum reasoning depth is fixed at $L$ due to the constraints of the depth of the model. Additionally, the maximum width of this tree is also limited, as the number of attention heads per layer is finite. For a reasoning tree $\mathcal{T}$ with $L$ layers, let the number of nodes in layer $i$ be denoted as $n_i$, and refer to the $j$-th node in the $i$-th layer as $u_{i,j}$. The retrieved document $d$ contains relevant information that can be leveraged to replace certain reasoning nodes with extracted document content. For instance, consider the query *Who is the actor playing Jason on General Hospital?'* (Yoran et al., 2023). In this scenario, there may be a node $u_{i,j}$ representing the information about *What is General Hospital?'*. If we provide a document containing detailed information about General Hospital, the computation for $u_{i,j}$ can effectively be replaced by extracting relevant information from that document.

**Definition 2.1** (Reasoning Tree). *For a reasoning tree $\mathcal{T}_L$ of $L$ layers, and there are $n_l$ nodes in each layer $l$, $n_{L-1} = 1$, all nodes in layer $l$ are connected to at least to one node in layer $l+1$. with probability $1 - q_l$, node $u_{l,i}$ are connected to $u_{l+1,j}$, and the isolated nodes are randomly connected to a node in case of useless nodes.*

In the reasoning tree, each node represents a calculation, with the output being some intermediate reasoning results. The connection between nodes $u_{i,j}$ and $u_{i-1,k}$ indicates that the calculation of $u_{i,j}$ relies on the output of $u_{i-1,k}$. The parameter $q_l$ reflects the sparsity of these connections. Consequently, for a node $u_{i,j}$, a document may contain relevant information, allowing the calculation for $u_{i,j}$ to be effectively replaced by information extracted from the document.

**Definition 2.2** (Retrieval). *For a reasoning tree $\mathcal{T}_L$, with probability $p_l$, the nodes in layer $l$ can be replaced by retrieved documents.*

It's important to recognize that processing documents also necessitates a certain depth of reasoning. However, extracting information from a document is fundamentally an information extraction process, whereas directly addressing a query is a purely inferential process. Therefore, we can assume that the cost of information extraction is lower than that of inference. This enables LLMs to leverage external information, thereby simplifying complex reasoning tasks.

**Assumption 2.3.** *If the document $d$ contains information about node $u$ at the $l$-th layer, then only $\lambda \cdot l$ layers are needed to extract the information from $d$, $\lambda < 1$.*

This assumption suggests that while processing documents requires some reasoning depth, it is generally shallower than the inference needed for directly addressing a query. Therefore, utilizing information from documents should be a more efficient solution. In this way, RAG can eliminate certain nodes and potentially enhance the expressive power of LLMs. However, this also indicates that RAG cannot completely minimize reasoning depth, as some layers are necessary for document processing. Consequently, for a reasoning tree $\mathcal{T}_L$, at least $\lambda \cdot L$ layers will be required to process the documents, even if we retrieve the documents directly lead to the answer.

The document not only simplifies the calculation of $u_{i,j}$ but also eliminates all nodes that are solely connected to $u_{i,j}$. These nodes contribute only to the reasoning of $u_{i,j}$, and since the information

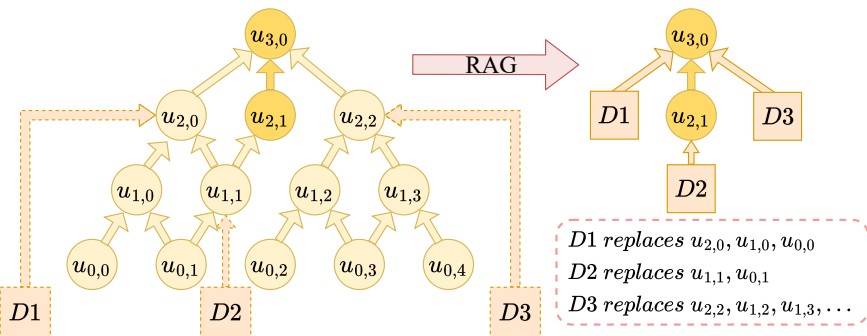

Figure 1: The reasoning tree of LLM. $u_{i,j}$ represents nodes, and $D1, D2, D3$ represents the document information.

of $u_{i,j}$ can be directly derived from the document, their reasoning is rendered unnecessary. Thus, retrieving a single document related to node $u_{i,j}$ can result in the reduction of multiple lower-layer nodes. This process is similar to fission reactions in nuclear weapons, where the reduction of one node triggers the reduction of many others. Consequently, if all nodes in the $l'$th layer are reduced by the Retrieval-Augmented Generation (RAG) method, any layer where $l \le l'$ can also be eliminated, effectively decreasing the overall reasoning depth.

As illustrated in Figure 1, the reasoning tree consists of 4 layers, and we have retrieved 3 documents that provide information for nodes $u_{2,0}$, $u_{1,1}$ and $u_{2,2}$, respectively. and with document $D1$, node $u_{1,0}$ can also be erased because it contributes only to $u_{2,0}$, and with $D2$, $u_{0,1}$ is not also necessary, the same applies to node $u_{1,2}$ and $u_{1,3}$ because of $D3$. Consequently, all 4 nodes in layer 1 can be either accessed or reduced through the documents, meaning that all nodes in layer 1 and 0 are unnecessary. This results in a reasoning depth of 2 instead of 4. Thus, with relevant documents, RAG can effectively reduce the reasoning complexity of the problem, enabling the LLM to solve problems of higher complexity.

We can observe that the elimination of a single node can significantly impact numerous nodes in shallower layers, much like a fission reaction. If this fission process can expand effectively, RAG could greatly enhance the reasoning capacity of LLMs. However, if the fission reaction halts at a certain threshold, the benefits may be limited. Therefore, to assess how many layers can be reduced by RAG, it is essential to determine whether this fission-like process can terminate. Understanding this dynamic is crucial for evaluating how RAG can improve reasoning capabilities and the overall efficiency of LLMs in complex problem-solving scenarios.

Consider a reasoning tree $\mathcal{T}_L$, let $t_{l+1}$ stands for the percentage of erased nodes in layer $l + 1$, then the probability of one node being erased in layer $l$ is affected by $t_{l+1}$ and the retrieval probability $p_l$, and the sparsity of the layer $q_l$, let $t_l = f(t_{l+1}, p_l, q_l)$, in the following of the paper, we will omit $p_l, q_l, t_{l+1}$, using simply $t_l = f(t)$.

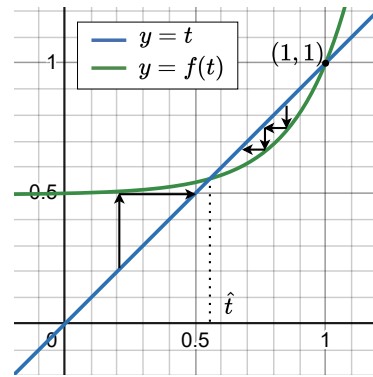

Figure 2: plot of $f(t)$ and $y = t$

The function $f(t)$ is expected to be increasing with respect to $t$, with $f(0) = p_l$ and $f(1) = 1$. Defining $g(t) = f(t) - t$, which represents growth in layer $l$, we can consider the existence of a point $\hat{t} \in (0, 1)$ where $g(\hat{t}) = 0$. If for $t > \hat{t}$, $g(t) < 0$, this indicates that the expected number of erased nodes will be smaller than in the previous layer, suggesting that the fission reaction will not propagate indefinitely but will instead reach a critical threshold. Beyond this point, the number

of nodes eliminated in the subsequent layer is expected to decrease compared to the current layer, thereby limiting the expansion of the fission reaction.

**Theorem 2.4** (Fission of RAG). *Let $t_l$ be the percentage of erased nodes in layer $l$, and let $n = n_{l+1}$. Then for layer $l$, if the retrival probability $p_l < 1 - \frac{1}{q_l^n - \ln q_l^n}$, then there exists $\hat{t} \in (0, 1)$ satisfying $g(\hat{t}) = 0$, and for $t > \hat{t}$, $g(t) < 0$.*

The proof is detailed in Appendix A.1. This theorem indicates that the erased nodes cannot propagate indefinitely like a fission reaction; rather, they will stop at a certain threshold. Specifically, when $p_l < 1 - \frac{1}{q_l^n - \ln q_l^n}$ and $t \geq \hat{t}$, it follows that $f(t) \leq t$. As illustrated in Figure 2, if $x \leq \hat{t}$, the probability of erasure increases, whereas for $x > \hat{t}$, it gradually decreases, stabilizing at $\hat{t}$.

The existence of $\hat{t}$ primarily depends on $p_l$ and $q_l^n$. In a regular reasoning tree, there are fewer nodes at the top and more nodes in shallower layers, resulting in tighter connections (smaller $q$ and $n$ at higher levels). Define $h(q, n) = 1 - \frac{1}{q_l^n - \ln q_l^n}$. Assuming $q \in [0.1, 0.8]$ and $n \in [2, 16]$, with $q_i - q_j = \eta(n_i - n_j)$, we find that $h(q, n) \geq 0.7$. This suggests that to sustain the fission reaction, we need to retrieve most intermediate reasoning results, which becomes challenging in higher layers, as we mainly retrieve information about basic concepts during reasoning. Consequently, this leads to the termination of the fission reaction in those upper layers. We set the maximal $n$ as 16 because Merrill & Sabharwal (2023) shows that two heads can simulate one reasoning node in the circuits, and for LLMs with 32 heads, the maximal $n$ is 16, and with larger $n$, $h(q, n)$ increases, making it harder for the fission reaction to continue.

Now we consider the scenario where the fission reaction converges to a certain point and calculate the probability of erasing a layer. As the probability of replacing a node approaches $\hat{t}_l$, we assume that the probability of replacing a node in layer $l$ is $\hat{t}_l$, i.e., $t_l = \hat{t}_l$. In Appendix A.2, we demonstrate through simulation that $t_l \approx \hat{t}_l$, which validates our assumption.

**Theorem 2.5.** *With $p_l < 1 - \frac{1}{q^n - n \ln q}$, if we want probability $\delta$ that all nodes in layer $l$ can be replaced by document information, we need $p_l \gtrsim 1 - \frac{\epsilon}{1 - q^{\epsilon n}}$, and $\epsilon \leq 1 - \sqrt[n]{\delta}$.*

The proof is presented in Appendix A.2. According to the theorem, $1 - \frac{\epsilon}{1 - q^{\epsilon n}}$ is decreasing with $\epsilon$, so for a feasible solution, we require a large $\epsilon$, i.e., a small $\sqrt[n]{\delta}$. Considering a similar scenario with $q \in [0.1, 0.8]$ and $n \in [2, 16]$, where $q_i - q_j = \eta(n_i - n_j)$, as shown in Figure 3, high-probability layer erasure is primarily achievable in lower layers, as we cannot retrieve much information from higher layers. However, we also observe that with a small probability, effectively erasing higher layers is possible, indicating that RAG can significantly enhance LLM reasoning. This suggests that if we are fortunate enough to retrieve documents that directly lead to the answer, we can substantially reduce reasoning complexity. Nonetheless, in typical scenarios, the assistance is limited since most retrieved documents contain only basic information.

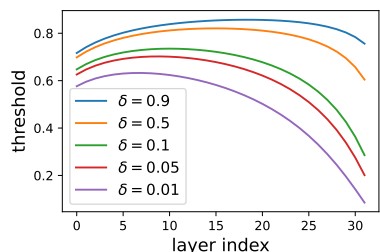

Figure 3: plot of the threshold, the y-axis is $1 - \frac{\epsilon}{1 - q^{\epsilon n}}$, x-axis is the layer and $\delta$ is the probability of replacing the whole layer

## 3 NOISE IN THE DOCUMENT

In real-world applications of RAG, Due to the limited capacity of the retriever, distractions frequently arise in the retrieved documents (Shi et al., 2023a; Wu et al., 2024). Moreover, even in relevant documents, the presence of noise information is common (Jiang et al., 2023b;a; Xu et al., 2024). So we cannot directly use the information from retrieved documents; further processing is necessary because those noise and distracting information can be mistakenly interpreted as relevant, significantly degrading the performance of RALM. In this section, we examine the extent to which distracting and noisy information impacts overall performance.

For a token $\boldsymbol{x}$, we assume that its embedding contains two distinct pieces of information: $\boldsymbol{w}$, which acts as the regressor to determine the relevance of the document to the query, and $\boldsymbol{s}$, representing other information useful for inference. Thus, we can express the relationship as $H(\boldsymbol{x}) = H(\boldsymbol{s}, \boldsymbol{w})$, where $H(\cdot)$ denotes entropy. Let $\boldsymbol{z}$ be the token embedding after passing through the transformer layers, and let $\boldsymbol{Z} = [\boldsymbol{z}_1^T, \ldots, \boldsymbol{z}_n^T]$, where $n$ is the sequence length. We use $\boldsymbol{Z}_r$ to denote the embeddings of the related tokens.

When generating new tokens, the token embedding is integrated into the inference process. Therefore, if more information about $\boldsymbol{w}$ is incorporated into $\boldsymbol{z}$ (resulting in a high $I(\boldsymbol{w}; \boldsymbol{z})$), the model can effectively recognize irrelevant tokens when conducting inference. Conversely, if the information is limited, the model is likely to be significantly affected by noise and distracting information in the input.

**Theorem 3.1** (Impact of Noise). *Given a set of input tokens, with $\delta$ percent of related and $1 - \delta$ percent of distractions. If we assume documents contribute the same, and LLM conducts the filtering before inference, then*

$$\mathcal{L}(\delta) - \mathcal{L}(1) \leq c_1 \sqrt{\left( \frac{(1-\delta)H(\boldsymbol{w}|\boldsymbol{z})}{\delta(I(\boldsymbol{w}; \boldsymbol{z}) + 1)} \right) \cdot H(\boldsymbol{Z}_r)} + c_2, \tag{1}$$

*where $\mathcal{L}(\delta)$ stands for the loss with $\delta$ percent of relevant documents and $\mathcal{L}(1)$ means the loss with all information related to the query, no distraction is involved. $\boldsymbol{z}$ is the embedding of documents, $H(\cdot)$ and $I(\cdot; \cdot)$ represents the entropy and mutual information. $c_1, c_2$ are constants and the value is shown in Appendix B.1.*

The proof is presented in Appendix B.1. From Equation 1, we see that the generalization error decreases as $\delta$ increases. This indicates that if not all information needed to distinguish irrelevant tokens is captured in the embedding, an increasing number of distracting documents will have negative affect on performance. The term $H(\boldsymbol{Z}_r)$ represents the amount of relevant token information utilized to infer the final result, suggesting that the greater the reliance of the LLM on documents, the more detrimental the impact of noise becomes. Furthermore, performance is primarily determined by $I(\boldsymbol{w}; \boldsymbol{z})$, which quantifies the information in the embedding relevant for assessing the document's relevance. Recent studies indicate that LLMs effectively perform information compression (Delétang et al., 2023; Huang et al., 2024; Yu et al., 2023), retaining only the information pertinent to the downstream task while discarding extraneous noise, specifically $H(\boldsymbol{z}|\boldsymbol{s})$. Consequently, only limited information about how to differentiate irrelevant documents is preserved, leading to diminished performance.

## 3.1 FINETUNING HELPS BUT NOT PERFECTLY

Clearly, noise in the documents negatively impacts the performance of LLMs. Some researchers focus on fine-tuning the model to better filter out irrelevant documents, thereby enhancing performance (Yoran et al., 2023; Zhang et al., 2024a; Jiang et al., 2023b). However, filtering out noise may require additional reasoning layers, which could hinder the reasoning ability of RALM. Therefore, we aim to explore whether we can incorporate the filtering process into the reasoning process, specifically addressing the question: **can we filter out irrelevant information while conducting the original inference?**

Let $\boldsymbol{r}$ represents the relevance of tokens, $r_i = 0$ means that token $\boldsymbol{x}_i$ is a noise token, otherwise the token is relevant. Let $attn(\boldsymbol{x}_i, \boldsymbol{x}_j) = (\boldsymbol{W}_q \boldsymbol{x}_i)^T \boldsymbol{W}_k \boldsymbol{x}_j$ represent the original attention layer of the LLM. And we assume that the desired self-attention function is:

$$\widehat{attn}(\boldsymbol{x}_i, \boldsymbol{x}_j) = \begin{cases} 0 & \text{if } r_j = 0, \\ (\boldsymbol{W}_q \boldsymbol{x}_i)^T \boldsymbol{W}_k \boldsymbol{x}_j & \text{else,} \end{cases} \tag{2}$$

The desired attention pattern should effectively exclude noise while preserving the original attention of relevant tokens. We assume that, in the presence of relevant tokens, the original model represents the optimal solution, as it is specifically trained to address the problem. Therefore, $\widehat{attn}$ can be considered the optimal response when confronted with noise, as it effectively filters out irrelevant tokens and utilizes the relevant information in the most efficient manner.

Finetuning the model involves adjusting its parameters, which allows the finetuned model to be expressed as $attn'(\boldsymbol{x}_i, \boldsymbol{x}_j) = \left((\boldsymbol{W}_q + \Delta \boldsymbol{W}_q)\boldsymbol{x}_i\right)^T (\boldsymbol{W}_k + \Delta \boldsymbol{W}_k)\, \boldsymbol{x}_j = \boldsymbol{x}_i^T (\boldsymbol{W} + \Delta \boldsymbol{W})\boldsymbol{x}_j$, where $\boldsymbol{W} = \boldsymbol{W}_q^T \boldsymbol{W}_k$ and $\Delta \boldsymbol{W}$ represents the adjustments. The critical question arises: can we finetune the model to approximate the optimal one, i.e., is there a $\Delta \boldsymbol{W}$ such that $\sigma(attn'(\boldsymbol{x}_i, \boldsymbol{x}_j)) \approx \sigma(\widehat{attn}(\boldsymbol{x}_i, \boldsymbol{x}_j))$? $\sigma(\cdot)$ represents the softmax.

**Theorem 3.2.** *if there exists* $attn'(\boldsymbol{x}_i, \boldsymbol{x}_j) = \boldsymbol{x}_i(\boldsymbol{W} + \Delta \boldsymbol{W})\boldsymbol{x}_j$, $\epsilon$ *approximates* $\widehat{attn}(\boldsymbol{x}_i, \boldsymbol{x}_j)$ *i.e.,* $(1 - \epsilon)\sigma(\widehat{attn}(x_i, x_j)) \leq \sigma(attn'(x_i, x_j)) \leq (1 + \epsilon)\sigma(\widehat{attn}(x_i, x_j))$, *then we need*

$$\delta \lesssim \ln \frac{1}{1 - \epsilon},$$

*where* $\delta = \max(\boldsymbol{x}_i^T \Delta \boldsymbol{W} \boldsymbol{x}_j) - \min(\boldsymbol{x}_i^T \Delta \boldsymbol{W} \boldsymbol{x}_j)$, *and* $x_j$ *is a relevant token.*

The proof is presented in Appendix B.2. To effectively fine-tune the model for filtering out irrelevant information in the attention matrix, we need $\delta \approx 0$. This implies that for all tokens to be retained, $\boldsymbol{x}_i^T \boldsymbol{W} \boldsymbol{x}_j$ must remain nearly constant. As a result, approximating the optimal solution proves to be quite challenging.

Since the self-attention layer struggles to filter out irrelevant information while maintaining optimal reasoning performance, a pertinent question arises: **can the feed-forward network connected to the self-attention layer effectively filter out mistakenly incorporated information?** Here, we denote the information necessary to assess the relevance of the involved tokens as $\hat{\boldsymbol{w}}$, so $H(\hat{\boldsymbol{w}}) = H(\boldsymbol{w}_{i1}, \boldsymbol{w}_{i2}, \ldots)$, where $\boldsymbol{w}_{ij}$ represents the relevance information of the token aggregated by self-attention and those tokens form the input to the feed forward layer.

**Theorem 3.3.** *For a Feed Forward Network* $f$ *and the input embedding* $v$ *contains* $1 - \delta$ *percent of noise information, assume the optimal function is* $\hat{f}(x)$ *which filter out the noise and finish the inference, then*

$$\Pr\left(\|f(x) - \hat{f}(x)\| > t'\right) > \frac{1}{C}\left(H(\boldsymbol{v}) - \delta \frac{I(\hat{\boldsymbol{w}}; \boldsymbol{v}) + 1}{H(\hat{\boldsymbol{w}})} \cdot I(\boldsymbol{s}; \boldsymbol{v})\right), \tag{3}$$

*where* $t$ *is a constant,* $t' = t + c_1 \sqrt{(1 - \delta)\frac{H(\hat{\boldsymbol{w}}|\boldsymbol{v}) - 1}{H(\hat{\boldsymbol{w}})} \cdot I(\boldsymbol{s}; \boldsymbol{v})} + c_2$. $H(\boldsymbol{v})$ *stands for all the information needed to conduct inference.* $C = \log \frac{|\mathcal{V}|}{N_{max}(t)}$ *is a constant and we show it in the Appendix.*

The proof is detailed in Appendix B.3. From the above theorem, it is evident that achieving better performance requires the input to incorporate information that effectively distinguishes irrelevant tokens (a large $I(\hat{\boldsymbol{w}}; \boldsymbol{v})$) and the necessary information for inference (a large $I(\boldsymbol{s}; \boldsymbol{v})$). Additionally, there should be minimal noise incorporated into the input (a large $\delta$). As previously noted, the self-attention module struggles to filter out irrelevant information while retaining relevant tokens, which makes it difficult to increase $\delta$. Furthermore, it is difficult to integrate additional information about $\hat{\boldsymbol{w}}$ (i.e., to increase $I(\hat{\boldsymbol{w}}; \boldsymbol{v})$) because the input to the MLP layer is given by $\boldsymbol{v} = \sum_j a_{i,j} \boldsymbol{x}_j \boldsymbol{W}_v$, resulting in $\boldsymbol{v}$ being an embedding of dimension $m$ and precision $p$. This limited dimensionality makes it challenging to encapsulate all the necessary information for filtering out irrelevant documents for all involved tokens. Consequently, the Feed Forward Layer is unlikely to achieve optimal performance when dealing with noisy input.

## 4 EXTRA LYAERS FOR FILTERING

Clearly, noise in the documents adversely affects the performance of large language models (LLMs), and filtering out this information without compromising the performance ability of the retrieval-augmented language model (RALM) is challenging. Additional layers are necessary for effectively filtering out irrelevant information. If we cannot resolve the filtering issue within a limited number of layers, the overall reasoning capability of the RALM may fall below that of standard reasoning models, resulting in retrieved documents failing to enhance the reasoning abilities of the LLM. Therefore, the crucial question is: **how many layers are required to effectively address the filtering problem?**

## 4.1 THE TRIPLE-WISE PROBLEM

For an input sequence $\boldsymbol{X} = [\boldsymbol{x}_0^T, \boldsymbol{x}_1^T, \ldots, \boldsymbol{x}_{n-1}^T]$, the vector $\boldsymbol{r}$ indicates the relevance of each token. Specifically, for each token $\boldsymbol{x}_i$, a relevance score of $r_i = 0$ signifies that the token is irrelevant to the query. It's crucial to note that calculating $r_i$ does not solely depend on the token embedding $\boldsymbol{x}_i$ and the query; rather, it may require the involvement of three or more tokens. For instance, in the query *"Alice is exhausted, but Bob is still very excited, showing no signs of fatigue. How does Bob feel?"*, the word *"exhausted"* acts as noise and should be excluded during inference. However, determining relevance necessitates considering *"Bob"* in the query alongside *"Alice"*, the subject of *"exhausted"*. Therefore, identifying the relevance of a token demands information from multiple tokens, yet self-attention computes relationships only between pairs, making it challenging to address this issue within a single transformer layer.

Consider a simple scenario in which three tokens are sufficient to determine the relevance of a given token. The task is to ascertain whether there exists information indicating that token $\boldsymbol{x}_i$ is noise relative to the query. Let $g(\boldsymbol{x}_i, \boldsymbol{x}_a, \boldsymbol{x}_b)$ be the function designed to identify whether tokens $\boldsymbol{x}_a$ and $\boldsymbol{x}_b$ suggest that token $\boldsymbol{x}_i$ is a relevant token, if $g(\boldsymbol{x}_i, \boldsymbol{x}_a, \boldsymbol{x}_b) = 0$, then $x_i$ is a noise token, otherwise it is relevant. Thus, the problem can be framed as follows:

$$r_i = \begin{cases} 0 & \text{if } \exists\, a, b \text{ s.t. } g(x_i, x_a, x_b) = 0, \\ 1 & \text{else.} \end{cases}$$

And we need a transformer based model $\mathcal{M}$ to calculate $\boldsymbol{r} = \mathcal{M}(\boldsymbol{X})$, and we can show that one layer of transformer can hardly complete the task.

**Theorem 4.1.** *For input documents of length $n$, and a constant $c$ if $mpH \leq c \cdot nH(\boldsymbol{w})/\log\log n$, then there is no one layer transformer $\mathcal{M}$ with embedding size $m$, precision $p$ and $H$ heads to judge the relevance of tokens.*

Consequently, a single layer of multi-head attention struggles to assess the relevance of a token. As noted in Conjecture 19 of Sanford et al. (2024), even multiple layers of multi-head attention may not effectively resolve this issue. The challenge arises from the triple-wise nature of the problem contrasted with the pair-wise nature of attention; the model can only evaluate a token's relevance when its embedding contains substantial information. For example, to assess the token *"fatigue"* in the sentence *"Alice is exhausted, but Bob is still very excited, showing no signs of fatigue. How does Bob feel?"*, the embedding must encompass information about its subject, *"Bob"*, as well as contextual details from phrases like *"no signs of"* and *"showing"*. Therefore, a significant amount of information must be incorporated into the embedding before any judgement can be made. However, a single layer of self-attention can only consider the input $\sum_j a_{i,j} \boldsymbol{x}_j$, which contains information up to $mp$, where $m$ represents the embedding dimension and $p$ indicates precision. The term $mp$ signifies the maximal entropy of the embedding, which means the maximal information the embedding can carry. This suggests that **multiple attention layers may be necessary to encompass all relevant information effectively.**

Furthermore, this method renders Assumption 2.3 unrealistic, as assessing relevance requires a deep understanding of the document, thereby increasing the number of layers necessary for effective information extraction. If we assume that $t$ layers are needed to evaluate relevance, then extracting information about node $u_{l,j}$ would require $\lambda \cdot l + t$ layers. This could potentially exceed $l$, and extracting information from documents may necessitate a greater reasoning depth than standard reasoning. Consequently, nodes located below layer $\frac{t}{1-\lambda}$ cannot access document information because their reasoning would be finished faster than information extraction of documents. Then the reasoning depth cannot be easily reduced, as it primarily occurs in the lower layers. Therefore, leveraging document information may not enhance reasoning ability; in fact, it could potentially degrade performance if the model becomes overly reliant on document information rather than employing vanilla reasoning.

## 4.2 SIMPLICIFY THE PROBLEM

Therefore, reducing the number of layers required for filtering is essential to enhancing the reasoning ability of retrieval-augmented language models (RALMs). In the retrieval-augmented generation

(RAG) setting, we can simplify the triple-wise problem. By precalculating information from the document and representing this summarized information as one or a few additional tokens (virtual prompt tokens), we can evaluate the relevance of a token using only the information from the token itself, the query, and the summarized document.

This approach simplifies the problem to:

$$r_i = \begin{cases} 0 & \text{if } \exists \, a, b \ s.t. \ g(x_i, x_v, x_b) = 0, \\ 1 & \text{else.} \end{cases}$$

where $x_v$ represents the virtual prompt token. By fixing the position of one token, we can utilize the first transformer layer to aggregate information of $x_i$ and $x_v$. In the second layer, the problem is actually a pair-wise one because $x_i$ already hold enough information about $x_v$, so $x_i$ and $x_b$ are enough to judge the relevance. This approach effectively reformulates the triple-wise problem into pair-wise one, which the transformer can more readily address within a single layer. This could be done because we fix the position of one information, then the triple-wise problem becomes two pair-wise problem. Otherwise, we need to incorporate plenty of information before final judgement, because we do not know which one could be useful when judging the relevance. With the virtual document token, we summarize those information about judging the relevance in a fixed position, and we can easily gather those information for final judgement

However, most large language models (LLMs) are trained in an autoregressive manner, meaning that a token cannot aggregate information from any tokens that appear later in the sequence; thus, $a_{i,j} = 0$ if $j > i$. Typically, the query is positioned after the document tokens, preventing these document tokens from assessing their relevance effectively. Consequently, the relevance judgement must occur during the calculation of the query embedding, complicating the assessment process and limiting the model's ability to utilize document information to judge the relevance.

Instead, if we position the query ahead of documents, then the relevance can be effectively calculated for document tokens. This arrangement enables the information of query to be effectively transferred to the document tokens for relevance judgement. Therefore, we can hypothesize that when there is noise in the document, placing the query at the beginning would help the judgement.

Considering the scenario where we aim to judge the relevance of the first token of the documents, with the relevance being determined by the first token of the query (a pair-wise situation), we can demonstrate that placing the query ahead is beneficial.

**Proposition 4.2.** *Using auto-regressive transformer to solve the pair-wise problem requires $mp \geq (n_{document} + 1)H(\boldsymbol{w})$. Putting the query ahead only requires $mp \geq 2H(\boldsymbol{w})$, where $n_{document}$ represents the number of documents.*

Table 1: The performance with query ahead

| | 2wiki | | hotpotqa | | musique | |
|---|---|---|---|---|---|---|
| | vanilla | reverse | vanilla | reverse | vanilla | reverse |
| LLama3 | 20.3 | 21 | 22.6 | 35.3 | 2.3 | 5.2 |
| Vicuna | 41.3 | 29.7 | 49.3 | 39.7 | 20.7 | 14.6 |
| Mistral | 39.6 | 51.3 | 52.4 | 55.4 | 14.3 | 19.7 |

This is primarily because documents cannot assess their relevance when the query is placed after them. In this scenario, the query must evaluate the relevance of all documents. However, if the query is positioned first, the documents can independently assess their own relevance. We conduct experiments on three models with multi-hop datasets, more details about the experiment is shown in Appendix D. As shown in Table 1, positioning the query ahead significantly enhances performance except for Vicuna. "vanilla" means that we placed the query after documents and "reverse" means that the query is placed both before and after the documents.

Apparently, different from LLama3 and Mistral, Vicuna performs worse when we place the query ahead of documents. We assume that this is mainly because of the finetuning of Vicuna. The data used for finetuning Vicuna is mainly collected from ShareGPT it contains 70K user-shared

multi-turn ChatGPT conversations, in this situation, previous conversation data is quite long and it functions similar to the document in RAG, they both provide information for current query. And when finetuning Vicuna, the previous conversation is placed before the current query (document before the query), so the model might overfit to the situation and when place the query ahead of documents, the model would perform bad.

## 5 DPROMPT TUNING

As stated in the previous section, effectively addressing this problem requires the use of virtual tokens at the beginning of the prompt to represent the document's information. To accomplish this, we propose training an additional model to extract information from the document and utilize the embeddings to encapsulate this information. We then append these virtual tokens to the front of the prompt before conducting inference.

It is important to note that the model used for information extraction does not impose any additional inference costs on the LLM, because the calculations are solely based on the document, This allows us to precalculate the embeddings and directly retrieve the information while retrieving documents, ensuring that there is no extra cost during inference.

Specifically, we fine-tune a BERT-base-uncased model with an additional multi-layer perceptron (MLP) to project the embeddings to the appropriate dimensions. Given an input prompt, we identify the relevant documents, encode them as virtual tokens, and then inject these tokens at the front of the prompt.

Additionally, we perform inference with the query both before and after the documents to optimize performance. And we put the virtual tokens represents the documents before the query for all tokens to effectively take advantage of the virtual token. We also employ LoRA fine-tuning methods to enhance the model's ability to leverage the information in the virtual document tokens.

Table 2: Performance of DPrompt tuning. Lora+Pro means LoRA+prompt tuning.

|          | vanilla | Lora | Prompt | Lora+Pro | Dprompt |
|----------|---------|------|--------|----------|---------|
| 2wiki    | 51.3    | 53.7 | 52.7   | 54.2     | **56.5** |
| hotpotqa | 55.4    | 55.7 | 55.6   | 55.9     | **56.7** |
| musique  | 19.7    | 20.3 | 20.4   | 21.2     | **21.8** |

We conduct experiments using multi-hop datasets including 2wikimultihopqa (Ho et al., 2020), hotpotqa (Yang et al., 2018) and musique (Trivedi et al., 2022), we utilize BM25 to extract documents, treating those retrieved documents that do not contain answer or middle reasoning result as noise (Jeong et al., 2024). The input to the LLM contains the evidence documents which contains middle reasoning results to the query and other three distracting documents, we finetune the model on those three datasets and show the performance in Table 2. We conduct experiments on Mistral-7b and evaluate the performance by accuracy, the accuracy means that the answer is contained in the generated output. More experimental details are shown in Appendix D.

Apparently, both LoRA and Prompt tuning can help the performance and DPrompt Tuning helps the most. DPrompt tuning achieves significant performance improvements across all three datasets. This indicates that adding virtual tokens at the beginning enhances the LLM's ability to distinguish irrelevant information and extracting the relevant ones. Also we conduct experiments to show how will the finetuned model performs on gold document only datasets in Appendix D.1.

## 6 CONCLUSION

In this paper, we highlight that while RAG can enhance the reasoning capabilities of LLMs, it often only expands a limited number of reasoning depth. However, noise within the documents can negatively affect performance, and processing these documents demands lots of reasoning depth due to its intrinsic triple-wise nature. To address this, we propose a method called DPrompt tuning which allows for effective resolution within few layers of the transformer.

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

# A EXPRESSIVE POWER OF RAG

## A.1 PROOF OF THEOREM 2.4

Assume there is a tree of $L$ layers and each layer has $n_l$ nodes, the top of the tree got only one node. There are connections between the layer and its upper layer, the probability of connection is $1 - q_l$, if node $u_{l,i}$ is not connected to any node, than we randomly sample a node and set the connection, the probability of no connection is obviously $q_l^{n_{l+1}}$

The connection can be regarded as reasoning paths, and RAG can ignore the reasoning path, directly give the answer corresponding to node $u_{l,i}$, $u_{l,i}$ stands for the $i^{th}$ node of the $l^{th}$ layer. Assume that node $u_{l,i}$ is retrived by RAG, then all the downstream nodes connect to and only to node $u_{l,i}$ can also be erased.

Assume in layer $l + 1$, $r_{l+1}$ nodes are erased, then for layer $l$, let $n = n_{l+1}$, the probability of nodes being erased is,

$$p_e = (1 - q_l^{r_{l+1}})q_l^{n-r_{l+1}} + q_l^n \frac{r_{l+1}}{n} \tag{4}$$

$1 - q_l^{r_{l+1}}$ means it connects to at least one of the $r_{l+1}$ nodes, and $q_l^{n-r_{l+1}}$ means that it is not connected to other nodes. So the first item $(1-q_l^{r_{l+1}})q_l^{n-r_{l+1}}$ means that when the node is connected to at least one of the $r_{l+1}$ nodes, and not connected to other nodes. And the second item $q_l^n \frac{r_{l+1}}{n}$ means that it is not connected to any of nodes, so the connection is randomly set, and the setted node is erased.

Also, RAG can retrieve some information about layer $l$, Assume that with probability $p_l$, RAG retrieves information about node $u_{l,i}$, then, the percentage of node erased by converge and RAG are $t_l$, $n$ stands for the number of nodes in layer $l + 1$

$$
\begin{aligned}
p &= p_e + p_l \cdot (1 - p_e) \\
&= \left( (1 - q_l^{r_{l+1}}) \cdot q_l^{n-r_{l+1}} + q_l^n \frac{r_{l+1}}{n} \right) + p_l \cdot \left( 1 - \left( (1 - q_l^{r_{l+1}}) \cdot q_l^{n-r_{l+1}} + q_l^n \frac{r_{l+1}}{n} \right) \right) \\
&= q_l^n \left( q_l^{-r_{l+1}} - 1 + \frac{r_{l+1}}{n} \right) + p_l - p_l q_l^n \left( q_l^{-r_{l+1}} - 1 + \frac{r_{l+1}}{n} \right) \\
&= (q_l^n - p_l q_l^n) \left( q_l^{-r_{l+1}} - 1 + \frac{r_{l+1}}{n} \right) + p_l
\end{aligned}
$$

$$\mathbb{E}(t_l) = p = (q_l^n - p_l q_l^n) \left( q_l^{-nt_{l+1}} - 1 + t_{l+1} \right) + p_l$$

$t_l$ stands for the percentage of nodes could be erased, consider a function $f(t)$ where

$$f(t) = (q_l^n - p_l q_l^n) \left( q_l^{-nt} - 1 + t \right) + p_l$$

$$f'(t) = (q_l^n - p_l q_l^n) \left( -nq_l^{-nt} \ln q_l + 1 \right)$$

Considering $g(t) = f(t) - t$ then,

$$
\begin{aligned}
g'(t) &= (q_l^n - p_l q_l^n) \left( -nq_l^{-nt} \ln q_l + 1 \right) - 1 \\
&= (1 - p_l)q_l^n(-nq_l^{-nt} \ln q_l + 1) - 1
\end{aligned}
$$

For simplicity, we use $p$ and $q$ instead of $p_l$ and $q_l$ When $g'(t) = 0$, we can dervie that:

$$(1 - p) \cdot q^n \cdot \left(-nq^{-nt} \ln q + 1\right) - 1 = 0$$

$$-(1 - p)q^n nq^{-nt} \ln q + (1 - p)q^n - 1 = 0$$

$$-(1 - p)q^n nq^{-nt} \ln q = 1 - (1 - p)q^n$$

$$q^{-nt} = \frac{1 - (1 - p)q^n}{(p - 1)q^n n \ln q}$$

$$-nt \ln q = \ln\left(1 - (1 - p)q^n\right) - \ln\left((p - 1)q^n n \ln q\right)$$

$$t = \frac{\ln\left((p - 1)q^n n \ln q\right) - \ln\left(1 - (1 - p)q^n\right)}{n \ln q}$$

$$t = \frac{n \ln q + \ln\left((p - 1)n \ln q\right) - \ln\left(1 - (1 - p)q^n\right)}{n \ln q}$$

$$t = 1 + \frac{\ln \frac{(p-1)n \ln q}{1-(1-p)q^n}}{n \ln q}$$

It is easy to see that $f'(t)$ is monotonically increasing with $t$ and $f'(t) > 0$, so when $t < 1 + \frac{\ln \frac{(p-1)n \ln q}{1-(1-p)q^n}}{n \ln q}$, $f'(t) < 1$ and reversely, when $t > 1 + \frac{\ln \frac{(p-1)n \ln q}{1-(1-p)q^n}}{n \ln q}$, $f'(t) > 1$.

So $g(t) = f(t) - t$ will decrease first and increase later, and $g(0) = f(0) - 0 = f(0) = p_l \geq 0$, and $g(1) = 0$. And the question is, is there a chance that $g(t) < 0$, which means that the nodes erased in layer $l$ is fewer that layer $l + 1$? In this case, the fission reaction can reach a bound, otherwise, the fission continues until it erase all nodes.

Apparently, if $\frac{\ln \frac{(p-1)n \ln q}{1-(1-p)q^n}}{n \ln q} >= 0$, then 1 will be the first zero-point of $g(t)$ because $g(t)$ monotonically decreases with $t$, so $g(t) >= 0$ with $t \in [0, 1]$. Otherwise, if $\frac{\ln \frac{(p-1)n \ln q}{1-(1-p)q^n}}{n \ln q} < 0$, then, $g(t)$ decreases first and increases, and let $g'(\hat{t}) = 0$, we have $g(\hat{t}) < 0$, then 1 will be the second zero-point, so $g(t)$ could be smaller than 0 with $t \in [0, 1]$.

Apparently $n \ln q < 0$, so with $\ln \frac{(p-1)n \ln q}{1-(1-p)q^n} > 0$, then $g(t)$ could be smaller than 0 with $t \in [0, 1]$.

$$\ln \frac{(p - 1)n \ln q}{1 - (1 - p)q^n} > 0$$

$$(p - 1)n \ln q - \left(1 - (1 - p)q^n\right) > 0$$

$$-(1 - p)n \ln q - 1 + (1 - p)q^n > 0$$

$$(1 - p)(q^n - n \ln q) > 1$$

$$1 - p > \frac{1}{q^n - n \ln q}$$

$$p < 1 - \frac{1}{q^n - n \ln q}$$

Therefore, when $p < 1 - \frac{1}{q^n - n \ln q}$, there exists $g(t) < 0$, and when $p >= 1 - \frac{1}{q^n - n \ln q}$, $g(t) >= 0$

When $g(t) < 0$, it means that $f(t) < t$, so for layer $l$, the number of erased nodes is smaller than layer $l + 1$. So the fission reaction fail, and there is a threshold for the number of erased number, the threshold is the first zero-point of $g(t)$.

**Theorem A.1** (fission of RAG). *For layer $l$, if the retrival probability $p_l < 1 - \frac{1}{q^n - n \ln q}$, then there exists $\hat{x} \in (0, 1)$ satisfying $g(\hat{x}) = 0$.*

Apparently, for lower levels of the tree, the connection is tighter and the retrieval is harder, therefore, it is impossible that $p_l >= 1 - \frac{1}{q^n - n \ln q}$ satisfies, so for lower levels, $t_l < t_{l+1}$. When the level gets higher, the retrieval becomes easier and the connection is sparser, so it is more likely to satisfy that $p_l >= 1 - \frac{1}{q^n - n \ln q}$. Therefore, with RAG, we can greatly erase the computation of first layers and even erase the whole layer which increases the expressive power of transformer.

## A.2 PROOF OF THEOREM 2.5

To assess whether $t$ converges to $\hat{t}$, we conducted a simulation experiment with 10 repetitions, where $p_l \sim N(0.8 - 0.06l, 0.01)$, $q_l \sim N(0.8 - 0.06l, 0.01)$, and $n_l \sim N(16 - 1.4l, 0.01)$, where $l$ represents the layer index. As depicted in figure 4, it is observed that $t$ is quite close to the corresponding $\hat{t}$, the critical point where the fission reaction stops, therefore, with high probability $t \approx \hat{t}$. Also we can notice that the variance in higher layers are quite large, this means that with probability that we happen to retrieve some information quite close to the answer, then we can erase the higher layers, but in normal cases, we can only reduce the lower layers. Also, the critical point decreases when the layer index increases, showing that we can replace more nodes in the lower levels. But due to the large number of nodes, replacing the whole layer is also a tricky problem.

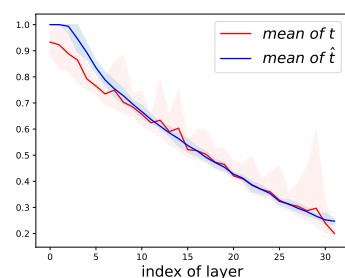

Figure 4: The simulated value of replacing probability

**Assumption A.2.** *The probability of erasing in layer $l$ satisfies $t_l \approx z_l$, $z_l$ is the first zero point of $g(t)$.*

For simplicity, we use $z$ instead of $z_l$, and $g(z) = 0$,

$$(q^n - pq^n)\left(q^{-nz} - 1 + z\right) + p - z = 0$$
$$-\left(q^{-nz} - 1 + z\right)q^n p + p + q^{n-nz} - q^n + q^n z - z = 0$$
$$\left(q^n - q^{n-nz} - q^n z + 1\right)p + q^{n-nz} - q^n + q^n z - z = 0$$
$$\left(q^n - q^{n-nz} - q^n z + 1\right)p = q^n - q^{n-nz} - q^n z + z$$
$$p = \frac{q^n - q^{n-nz} - q^n z + z}{q^n - q^{n-nz} - q^n z + 1}$$

Assume $\xi = q^{n-nz} + q^n z - q^n$, then $p = \frac{z - \xi}{1 - \xi}$.

If we need $t_l \geq \sqrt[n]{\delta}$, then,

$$z_l \geq \sqrt[n]{\delta}$$
$$p \geq \frac{\sqrt[n]{\delta} - \xi}{1 - \xi}$$

Assume $z = 1 - \epsilon$, then, $\epsilon \leq 1 - \sqrt[n]{\delta}$, $\xi = q^{n(1-z)} + q^n z - q^n = q^{n\epsilon} - \epsilon q^n$.

$$p = \frac{1 - \epsilon - q^{\epsilon k} + \epsilon q^n}{1 - q^{\epsilon k} + \epsilon q^n} = 1 - \frac{\epsilon}{1 - q^{\epsilon k} + \epsilon q^n} \approx 1 - \frac{\epsilon}{1 - q^{\epsilon k}}$$

**Theorem A.3.** *With $p_l < 1 - \frac{1}{q^n - n \ln q}$, if we want probability $\delta$ that layer $l$ can be totally covered i.e., $r_l^n \geq \delta$, we need $p_l \gtrsim 1 - \frac{\epsilon}{1 - q^{\epsilon n}}$, and $\epsilon \leq 1 - \sqrt[n]{\delta}$.*

## B THE IMPACT OF NOISE

### B.1 PROOF OF THEOREM 3.1

**Assumption B.1.** *The model conducts filtering before inference, and those information regarded as noise would not be used during inference. When there are $\delta$ percent of relevant tokens and $1 - \delta$ percent of noise information are included in the inference, let $\boldsymbol{X}_r$ and $\boldsymbol{X}_n$ represents the*

*relevant tokens and noise, $\boldsymbol{X}$ represents the input to the inference, then $I(\boldsymbol{X}_r; \boldsymbol{X}) = \delta H(\boldsymbol{X})$ and $I(\boldsymbol{X}_n; \boldsymbol{X}) = (1 - \delta)H(\boldsymbol{X})$*

Assume that there are $\delta$ percent of tokens are actually related to the query, and other $1 - \delta$ percent tokens are distractions. In addition, with probability $p_e$ the model mistakenly classify is the document relevant or not. Then, after filtering, $\delta \cdot (1 - p_e)$ percent of relevant tokens and $(1 - \delta) \cdot p_e$ distractions are left in the input. Then the percentage of distraction is,

$$\alpha = \frac{(1-\delta) \cdot p_e}{(1-\delta) \cdot p_e + \delta \cdot (1-p_e)} = \frac{(1-\delta)p_e}{p_e + \delta - 2\delta p_e} = \frac{1-\delta}{\frac{\delta}{p_e} - 2\delta + 1}$$

Based on Fano's inequality, $p_e \geq \frac{H(\boldsymbol{w}|\boldsymbol{z})-1}{H(\boldsymbol{w})}$, $\boldsymbol{z}$ is the embedding obtained by transformer, and we assume that $\boldsymbol{s}$ is the information we need to extract from document for further generation, $\boldsymbol{w}$ is the information to identify the relevance of tokens.

$$H(\boldsymbol{w}|\boldsymbol{z}) = H(\boldsymbol{w}) - I(\boldsymbol{w}; \boldsymbol{z})$$

$$p_e \geq \frac{H(\boldsymbol{w}) - I(\boldsymbol{w}; \boldsymbol{z}) - 1}{H(\boldsymbol{w})}$$

$$\alpha \geq \frac{1-\delta}{\frac{\delta H(\boldsymbol{w})}{H(\boldsymbol{w})-I(\boldsymbol{w};\boldsymbol{z})-1} - 2\delta + 1}$$

**Theorem B.2** (Theorem 1 of Kawaguchi et al. (2023)). *Let $l \in \{1, \ldots, D\}$. Suppose that $\phi_l^s$ is fixed independently of the training dataset $s$. Then, for any $\delta > 0$, with probability at least $1 - \delta$, the following holds:*

$$\Delta(s) \leq G_3^l \sqrt{\frac{I(X; Z_l|Y) \ln(2) + G_2^l}{n}} + \frac{G_1^l(0)}{\sqrt{n}},$$

*where $G_1^l(0) = \mathcal{O}(1)$, $G_2^l = \mathcal{O}(1)$, and $G_3^l = \mathcal{O}(1)$, as $n \to \infty$.*

The theorem above regard the first $l$ layers of a model as a encoder, and the rest as decoder. In LLM, we encode the token embeddings one by one and then decode from those encoded embedding to generate new tokens. In this way, $Z_l$ is actually the output embeddings of tokens, and $I(X; Z|Y) = H(\boldsymbol{Z}_d) = \frac{\alpha}{1-\alpha} H(\boldsymbol{Z}_r)$, where $H(\boldsymbol{Z}_d)$ stands for the entropy of those distracting tokens and $H(\boldsymbol{Z}_r)$ stands for the entropy of those relevant tokens.

$$\begin{aligned} \frac{\alpha}{1-\alpha} &= \frac{(1-\delta)p_e}{p_e + \delta - 2\delta p_e} \cdot \frac{p_e + \delta - 2\delta p_e}{p_e + \delta - 2\delta p_e - (1-\delta)p_e} \\ &= \frac{(1-\delta)p_e}{p_e + \delta - 2\delta p_e - (1-\delta)p_e} \\ &= \frac{(1-\delta)p_e}{\delta - \delta p_e} = \frac{1-\delta}{\frac{\delta}{p_e} - \delta} \end{aligned}$$

then, if we assume that the model can perfectly take advantage of existing information

$$\begin{aligned} I(X; Z|Y) &= \frac{\alpha}{1-\alpha} \cdot H(\boldsymbol{Z}_r) \\ &= \left( \frac{(1-\delta)(H(\boldsymbol{w}|\boldsymbol{z})-1)}{\delta(I(\boldsymbol{w};\boldsymbol{z})+1)} \right) \cdot H(\boldsymbol{Z}_r) \end{aligned} \quad (5)$$

Using Theorem B.2, we can observe that,

$$\Delta(s) \leq G_3 \sqrt{\frac{I(X;Z|Y)\ln(2) + G_2}{n}} + \frac{G_1(0)}{\sqrt{n}}$$

$$\leq G_3 \sqrt{\frac{I(X;Z|Y)\ln(2)}{n}} + \sqrt{\frac{G_2}{n}} + \frac{G_1(0)}{\sqrt{n}}$$

$$= c_1 \sqrt{I(X;Z|Y)} + c_2 \tag{6}$$

$$= c_1 \sqrt{\left(\frac{(1-\delta)(H(\boldsymbol{w}|\boldsymbol{z}) - 1)}{\delta(I(\boldsymbol{w};\boldsymbol{z}) + 1)}\right) \cdot H(\boldsymbol{Z}_r)} + c_2$$

$$\leq c_1 \sqrt{\left(\frac{(1-\delta)H(\boldsymbol{w}|\boldsymbol{z})}{\delta(I(\boldsymbol{w};\boldsymbol{z}) + 1)}\right) \cdot H(\boldsymbol{Z}_r)} + c_2$$

where $c_1 = G_3\sqrt{\frac{\ln 2}{n}}$, $c_2 = \sqrt{\frac{G_2}{n}} + \frac{G_1(0)}{\sqrt{n}}$

$$\mathcal{L}(\delta) - \mathcal{L}(1) \leq c_1 \sqrt{\left(\frac{(1-\delta)H(\boldsymbol{w}|\boldsymbol{z})}{\delta(I(\boldsymbol{w};\boldsymbol{z}) + 1)}\right) \cdot H(\boldsymbol{Z}_r)} + c_2, \tag{7}$$

**Theorem B.3** (Impact of Distractions)**.** *Given a set of documents, with $\delta$ percent of related and $1-\delta$ percent of distractions. If we assume documents contribute the same, and we ask the LLM to ignore the irrelevant documents, then,*

$$\mathcal{L}(\delta) - \mathcal{L}(1) \leq c_1 \sqrt{\left(\frac{(1-\delta)H(\boldsymbol{w}|\boldsymbol{z})}{\delta(I(\boldsymbol{w};\boldsymbol{z}) + 1)}\right) \cdot H(\boldsymbol{Z}_r)} + c_2, \tag{8}$$

*where $\mathcal{L}(\delta)$ stands for the loss with $(1 - \delta)$ distractions and $\mathcal{L}(1)$ means the training loss as in the training stage, no distraction is involved. $I(\boldsymbol{Z}_d; \boldsymbol{Y})$ is the mutual information between the embedding of related documents and the final embedding, this shows how much does LLM rely on documents during inference.*

## B.2 PROOF OF THEOREM 3.2

Let $attn(x_i) = (\boldsymbol{W}_q \boldsymbol{x}_i)^T \boldsymbol{W}_k \boldsymbol{X}_{:i}$ be the original attention layer of LLM, and $attn'(x_i) = ((\boldsymbol{W}_q + \Delta\boldsymbol{W}_q)\boldsymbol{x}_i)^T (\boldsymbol{W}_k + \Delta\boldsymbol{W}_k)\boldsymbol{X}_{:i}$ be the finetuned one and $\widehat{attn}$ be desired function, can we finetune the model to be $\widehat{attn}$?

So we need

$$softmax(attn'(\boldsymbol{X}))[i] \approx \begin{cases} 0 & \text{if } \exists b \ s.t. \ g_1(x_i, x_b) = 0 \\ softmax(attn(\boldsymbol{X}_r))[i] & \text{else} \end{cases}$$

where $\boldsymbol{X}_r$ means those related tokens. let $\boldsymbol{A}_{i,j} = attn(\boldsymbol{x}_i, \boldsymbol{x}_j) = (\boldsymbol{W}_q\boldsymbol{x}_i)^T \boldsymbol{W}_k \boldsymbol{x}_j = \boldsymbol{x}_i^T \boldsymbol{W} \boldsymbol{x}_j$

$$\begin{aligned} attn'(\boldsymbol{x}_i, \boldsymbol{x}_j) &= ((\boldsymbol{W}_q + \Delta\boldsymbol{W}_q)\boldsymbol{x}_i)^T (\boldsymbol{W}_k + \Delta\boldsymbol{W}_k)\boldsymbol{x}_j \\ &= \boldsymbol{x}_i(\boldsymbol{W} + \Delta\boldsymbol{W})\boldsymbol{x}_j \\ &= \boldsymbol{x}_i\boldsymbol{W}\boldsymbol{x}_j + \boldsymbol{x}_i\Delta\boldsymbol{W}\boldsymbol{x}_j \end{aligned}$$

where $\Delta\boldsymbol{W} = \Delta\boldsymbol{W}_q\boldsymbol{W}_k + \boldsymbol{W}_q\Delta\boldsymbol{W}_k + \Delta\boldsymbol{W}_q\Delta\boldsymbol{W}_k$

if there exists $\Delta\boldsymbol{W}_q'$, $\Delta\boldsymbol{W}_k'$ satisfying

$$\boldsymbol{x}_i\Delta\boldsymbol{W}\boldsymbol{x}_j \begin{cases} \leq c_l & \text{if } x_j \text{ is noise} \\ \in [c_h, c_h + \delta] & \text{else} \end{cases} \tag{9}$$

if $\boldsymbol{x}_j$ is a noise token and $\boldsymbol{x}_i \Delta \boldsymbol{W} \boldsymbol{x}_j = c_l$, then

$$softmax(attn'(\boldsymbol{x}_i))[j] - 0 = softmax(A_{i,:} + x_i \Delta \boldsymbol{W} \boldsymbol{X})[j]$$

$$= \frac{\exp(A_{i,j} + x_i \Delta \boldsymbol{W} x_j)}{\sum_k \exp(A_{i,k} + \boldsymbol{x}_i \Delta \boldsymbol{W} \boldsymbol{x}_k)}$$

$$= \frac{\exp(A_{i,j} + c_l)}{\sum_k \exp(A_{i,k} + \boldsymbol{x}_i \Delta \boldsymbol{W} \boldsymbol{x}_k)}$$

$$= \frac{\exp(A_{i,j} + c_l - c_h)}{\sum_k \exp(A_{i,k} + \boldsymbol{x}_i \Delta \boldsymbol{W} \boldsymbol{x}_k - c_h)}$$

if we need $softmax(attn'(\boldsymbol{x}_i))[j] - 0 \leq \epsilon^2$, due to the optimal value is 0, so to $\epsilon$ approximate the value, we use $\epsilon^2$, and let $A_{i,j} = \max(A_{i,:})$, then

$$\frac{\exp(A_{i,j} + c_l - c_h)}{\sum_k \exp(A_{i,k} + \boldsymbol{x}_i \Delta \boldsymbol{W} \boldsymbol{x}_k - c_h)} \leq softmax(attn'(\boldsymbol{x}_i))[j] - 0 \leq \epsilon^2$$

$$\exp(A_{i,j} + c_l - c_h) \leq \epsilon^2 \sum_k \exp(A_{i,k} + \boldsymbol{x}_i \Delta \boldsymbol{W} \boldsymbol{x}_k - c_h)$$

$$c_l - c_h \leq \ln\left(\epsilon^2 \sum_k \exp(A_{i,k} + \boldsymbol{x}_i \Delta \boldsymbol{W} \boldsymbol{x}_k - c_h)\right) - A_{i,j}$$

$$c_l - c_h \leq \ln\left(\epsilon^2 n \exp(A_{i,j} + \delta)\right) - A_{i,j}$$

$$c_l - c_h \leq \delta + \ln \epsilon^2 n$$

else if $\boldsymbol{x}_j$ is a relevant token, let $c = c_h$, and $\sum_{k'} \exp(A_{i,k'})$ denotes the summation of all relevant tokens, we consider a simple case where $\boldsymbol{x}_i \Delta \boldsymbol{W} \boldsymbol{x}_j = c_h$, and for one token $\boldsymbol{x}_{k1}$, we have $\boldsymbol{x}_i \Delta \boldsymbol{W} \boldsymbol{x}_{k1} = c_h + \delta$, and for all other relevant tokens we have $\boldsymbol{x}_i \Delta \boldsymbol{W} \boldsymbol{x}_k = c_h$

$$softmax(\hat{attn}(\boldsymbol{x}_i))[j] - softmax(attn'(\boldsymbol{x}_i))[j]$$

$$= \frac{\exp(A_{i,j})}{\sum_{k'} \exp(A_{i,k'})} - \frac{\exp(A_{i,j} + x_i \Delta \boldsymbol{W} x_j)}{\sum_k \exp(A_{i,k} + x_i \Delta \boldsymbol{W} x_k)}$$

$$= \frac{\exp(A_{i,j})}{\sum_{k'} \exp(A_{i,k'})} - \frac{\exp(A_{i,j} + x_i \Delta \boldsymbol{W} x_j - c)}{\sum_k \exp(A_{i,k} + x_i \Delta \boldsymbol{W} x_k - c)} \tag{10}$$

$$= \frac{\exp(A_{i,j})}{\sum_{k'} \exp(A_{i,k'})} - \frac{\exp(A_{i,j})}{\sum_k \exp(A_{i,k} + x_i \Delta \boldsymbol{W} x_k - c)}$$

considering $softmax(\hat{attn}(\boldsymbol{x}_i))[j] - softmax(attn'(\boldsymbol{x}_i))[j] = c(\frac{1}{a} - \frac{1}{b}) \leq \epsilon \frac{c}{a}$, $c = \exp(A_{i,j})$, $a = \sum_{k'} \exp(A_{i,k'})$, $b = \sum_k \exp(A_{i,k} + x_i \Delta \boldsymbol{W} x_k - c)$

$$b - a \leq \epsilon b$$

$$(1 - \epsilon)b \leq a$$

$$b \leq \frac{a}{1 - \epsilon}$$

$$\sum_k \exp(A_{i,k} + x_i \Delta \boldsymbol{W} x_k - c) \leq \frac{\sum_{k'} \exp(A_{i,k'})}{1 - \epsilon}$$

$$\sum_{k'} \exp(A_{i,k'} + x_i \Delta \boldsymbol{W} x_k - c) \leq \frac{\sum_{k'} \exp(A_{i,k'})}{1 - \epsilon}$$

$$\sum_{k'-k1} \exp(A_{i,k'}) + \exp(A_{i,k1} + \delta) \leq \frac{\sum_{k'} \exp(A_{i,k'})}{1 - \epsilon}$$

$$\exp(A_{i,k1} + \delta) \leq \frac{\epsilon}{1 - \epsilon} \sum_{k'-k} \exp(A_{i,k'}) + \frac{\exp(A_{i,k1})}{1 - \epsilon}$$

$$\delta \leq \ln\left(\frac{\epsilon}{1-\epsilon}\sum_{k'-k}\exp(A_{i,k'}) + \frac{\exp(A_{i,k1})}{1-\epsilon}\right) - A_{i,k1}$$

Considering the case that $\frac{\epsilon}{1-\epsilon}\sum_{k'-k}\exp(A_{i,k'}) \approx 0$, then we need $\delta \lesssim \ln\frac{1}{1-\epsilon}$

**Theorem B.4.** *if there exists* $attn'(x_i, x_j) = \boldsymbol{x}_i(\boldsymbol{W} + \Delta\boldsymbol{W})\boldsymbol{x}_j$ *satisfying* $attn'(x_i, x_j) - \widehat{attn}(x_i, x_j) \leq \epsilon$, *then we need*

$$\delta \lesssim \ln\frac{1}{1-\epsilon}$$

$$c_l - c_h \leq \delta + \ln\epsilon^2 n$$

### B.3 PROOF OF THEOREM 3.3

In the following, we make use of the quantities

$$N_{max}(t) = \max_{\hat{v}\in\hat{\mathcal{V}}} N_{\hat{v}}(t), \qquad N_{min}(t) = \min_{\hat{v}\in\hat{\mathcal{V}}} N_{\hat{v}}(t),$$

where

$$N_{\hat{v}}(t) = \sum_{v\in\mathcal{V}} \mathbb{1}\{d(v, \hat{v}) \leq t\}$$

counts the number of $v \in \mathcal{V}$ within a "distance" $t$ of $\hat{v} \in \hat{\mathcal{V}}$.

**Theorem B.5.** (Fano's inequality with approximate recovery in Scarlett & Cevher (2019)) *For any random variables* $v, \hat{v}$ *on the finite alphabets* $\mathcal{V}, \hat{\mathcal{V}}$, *we have*

$$P_e(t) \geq \frac{H(\boldsymbol{v}|\hat{\boldsymbol{v}}) - 1}{\log\frac{|\mathcal{V}|}{N_{max}(t)}}. \tag{11}$$

$P_e(t) = \Pr(||\boldsymbol{z} - \hat{\boldsymbol{z}}|| > t)$, *where* $\boldsymbol{z}$ *is inferenced by some function and the input is* $\boldsymbol{v}$.

Assume the input to the feed forward layer $\boldsymbol{v}$, the first $l$ layers are used to identify the relevance and the last few layers are used for inference. The output of the first $l$ layers are $\boldsymbol{v}_l$, and the embedding is used to conduct inference and get the result $\boldsymbol{z}$. So we can say that with probability $p \geq \frac{H(\boldsymbol{v}_l|\hat{\boldsymbol{v}})-1}{\log\frac{|\hat{\mathcal{S}}|}{N_{max}(t)}}$ the resulting embedding fails to be $t$ close to the original one *i.e.,* $d(\boldsymbol{z}, \hat{\boldsymbol{z}}) \leq t$, where $\hat{\boldsymbol{v}}$ stands for the optimal input where all irrelevant information is filtered and all the related information is contained and $\hat{\boldsymbol{z}}$ is the corresponding output..

Assume that $\hat{\boldsymbol{w}}$ are equally distributed to each token which satisfies $I(\boldsymbol{w}_i; \boldsymbol{v}) = I(\boldsymbol{w}_j; \boldsymbol{v})$, therefore, for each document, the model holds the same probability of mistakenly identify its relevance. Let $p_{we}$ stands for the error probability of identify noise tokens, and $\delta$ be the percentage of relevant tokens. With probability $\delta p_{we}$, the relevant token is mistakenly regarded as irrelevant, and with probability $(1-\delta)p_{we}$ the irrelevant token is mistakenly regarded as relevant. So there are $\frac{\delta(1-p_{we})}{\delta(1-p_{we})+(1-\delta)p_{we}}$ percent of information about the relevant ones. Also $p_{we}$ percent of relevant information and $1-p_{we}$ percent of irrelevant information are discarded, then $I(\boldsymbol{s}; \boldsymbol{v}_l) = ((1-p_{we})\cdot\delta + p_{we}\cdot(1-\delta))I(\boldsymbol{s}; \boldsymbol{v})$ are the left information about inference, among these, $\frac{\delta(1-p_{we})}{\delta(1-p_{we})+(1-\delta)p_{we}}$ are acutally related information, others are noise information.

In this way,

$$\begin{aligned} I(\boldsymbol{v}_l; \hat{\boldsymbol{v}}) &= I(\boldsymbol{v}_l; \boldsymbol{s}) \\ &= \frac{\delta(1-p_{we})}{\delta(1-p_{we})+(1-\delta)p_{we}}\cdot((1-p_{we})\cdot\delta + p_{we}\cdot(1-\delta))I(\boldsymbol{s}; \boldsymbol{v}) \\ &= \delta(1-p_{we})\cdot I(\boldsymbol{s}; \boldsymbol{v}) \\ &\leq \delta(1-\frac{H(\boldsymbol{w}|\boldsymbol{v}-1)}{H(\boldsymbol{w})})\cdot I(\boldsymbol{s}; \boldsymbol{v}) \\ &= \delta(\frac{I(\boldsymbol{w}; \boldsymbol{v})+1}{H(\boldsymbol{w})})\cdot I(\boldsymbol{s}; \boldsymbol{v}) \end{aligned}$$

Therefore,

$$P_e(t) \geq \frac{H(\boldsymbol{v}|\hat{\boldsymbol{v}}) - 1}{\log \frac{|\mathcal{V}|}{N_{max}(t)}} = \frac{H(\boldsymbol{v}) - I(\boldsymbol{v}; \hat{\boldsymbol{v}})}{\log \frac{|\mathcal{V}|}{N_{max}(t)}}$$

$$\geq \frac{H(\boldsymbol{v}) - g_1(\delta, I(\boldsymbol{w}; \boldsymbol{v})) \cdot I(\boldsymbol{s}; \boldsymbol{v})}{\log \frac{|\mathcal{V}|}{N_{max}(t)}}$$

where $g_1(\delta, I(\boldsymbol{w}; \boldsymbol{v})) = \delta(\frac{I(\boldsymbol{w};\boldsymbol{v})+1}{H(\boldsymbol{w})})$

So when there is no noise, the inference can be conducted based on tose information, then we have

$$\Pr\left(||z - \hat{z}|| > t\right) \geq \frac{H(\boldsymbol{v}) - g_1(\delta, I(\boldsymbol{w}; \boldsymbol{v})) \cdot I(\boldsymbol{s}; \boldsymbol{v})}{\log \frac{|\mathcal{V}|}{N_{max}(t)}}$$

$$\Pr\left(||z - \hat{z}|| \leq t\right) \leq 1 - \frac{H(\boldsymbol{v}) - g_1(\delta, I(\boldsymbol{w}; \boldsymbol{v})) \cdot I(\boldsymbol{s}; \boldsymbol{v})}{\log \frac{|\mathcal{V}|}{N_{max}(t)}}$$

Considering the noise in the embedding of $\boldsymbol{v}_l$, and the noise would have negative impact on the inference.

Also the extra noise information contained in $\boldsymbol{v}_l$ is

$$I(\boldsymbol{v}^-; \boldsymbol{v}_l) = \frac{(1-\delta)p_{we}}{\delta(1-p_{we}) + (1-\delta)p_{we}} \cdot \cdot \left((1-p_{we}) \cdot \delta + p_{we} \cdot (1-\delta)\right) I(\boldsymbol{s}; \boldsymbol{v})$$

$$= (1-\delta)p_{we} \cdot I(\boldsymbol{s}; \boldsymbol{v})$$

$$= (1-\delta)\frac{H(\boldsymbol{w}|\boldsymbol{v}) - 1}{H(\boldsymbol{w})} \cdot I(\boldsymbol{s}; \boldsymbol{v})$$

**Theorem B.6** (Theorem 2 of Kawaguchi et al. (2023)). *Let $\mathcal{D} \subseteq \{1, 2, \ldots, D+1\}$. Then, for any $\delta > 0$, with probability at least $1 - \delta$ over the training set s, the following generalization bound holds:*

$$\Delta(s) \leq \min_{l \in \mathcal{D}} Q_l, \tag{12}$$

*where for $l \leq D$,*

$$Q_l = G_3^l \sqrt{\frac{\left(I(X; Z_l^s|Y) + I(\phi_l^S; S)\right)\ln(2) + \widehat{\mathcal{G}}_2^l}{n}} + \frac{G_1^l(\zeta)}{\sqrt{n}};$$

*and for $l = D+1$,*

$$Q_l = \mathcal{R}(f^s)\sqrt{\frac{I(\phi_l^S; S)\ln(2) + \check{\mathcal{G}}_2^l}{2n}},$$

*Here, $S \sim \mathcal{P}^{\otimes n}$, $G_1^l(\zeta) = \hat{\mathcal{O}}(\sqrt{I(\phi_l^S; S) + 1})$, $\widehat{\mathcal{G}}_2^l = \hat{\mathcal{O}}(1)$, $\check{\mathcal{G}}_2^l = \hat{\mathcal{O}}(1)$, and $G_3^l = \hat{\mathcal{O}}(1)$ as $n \to \infty$. The formulas of $G_1^l(\zeta)$, $\widehat{\mathcal{G}}_2^l$, $\check{\mathcal{G}}_2^l$, and $G_3^l$ are given in Appendix.*

using $||f(x) - \hat{f}(x)||$ as the loss function, then we have that

$$\mathcal{L} - \hat{\mathcal{L}} \leq G_3^l \sqrt{\frac{\left(I(X; Z_l^s|Y) + I(\phi_l^S; S)\right)\ln(2) + \widehat{\mathcal{G}}_2^l}{n}} + \frac{G_1^l(\zeta)}{\sqrt{n}}$$

$$= c_1\sqrt{I(\boldsymbol{v}^-|\boldsymbol{v}_l) + I(\phi_l^S; S)} + c_3 \tag{13}$$

$$\leq c_1\sqrt{I(\boldsymbol{v}^-|\boldsymbol{v}_l)} + c_1\sqrt{I(\phi_l^S; S)} + c_3$$

$$\leq c_1\sqrt{I(\boldsymbol{v}^-|\boldsymbol{v}_l)} + c_2$$

where $c_2 = c_3 + c_1\sqrt{I(\phi_l^S; S)}$ Therefore,

$$\Pr\left(||f(x) - \hat{f}(x)|| \leq t + c_1\sqrt{I(\boldsymbol{v}^-|\boldsymbol{v}_l)+} + c_2\right) \leq 1 - \frac{H(\boldsymbol{v}) - g_1(\delta, I(\boldsymbol{w}; \boldsymbol{v})) \cdot I(\boldsymbol{s}; \boldsymbol{v})}{\log \frac{|\mathcal{V}|}{N_{max}(t)}} \tag{14}$$

with $I(\boldsymbol{v}^-; \boldsymbol{v}_l) = (1-\delta)\frac{H(\boldsymbol{w}|\boldsymbol{v})-1}{H(\boldsymbol{w})} \cdot I(\boldsymbol{s}; \boldsymbol{v}) = g_2(\delta, I(\boldsymbol{w}; \boldsymbol{v})) \cdot I(\boldsymbol{s}; \boldsymbol{v})$.

$$\Pr\left(\|f(x) - \hat{f}(x)\| > t + c_1\sqrt{g_2(\delta, I(\boldsymbol{w}; \boldsymbol{v})) \cdot I(\boldsymbol{s}; \boldsymbol{v})+} + c_2\right)$$
$$> \frac{H(\boldsymbol{v}) - g_1(\delta, I(\boldsymbol{w}; \boldsymbol{v})) \cdot I(\boldsymbol{s}; \boldsymbol{v})}{\log\frac{|\mathcal{V}|}{N_{max}(t)}} \tag{15}$$

**Theorem B.7.** *For a Feed Forward Network $f$ and the input $x$ contains $1 - \delta$ percent of noise information, assume the optimal function is $\hat{f}(x)$ which filter out the noise and finish the inference, then*

$$\Pr\left(\|f(x) - \hat{f}(x)\| > t'\right) > \frac{H(\boldsymbol{v}) - g_1(\delta, I(\boldsymbol{w}; \boldsymbol{v})) \cdot I(\boldsymbol{s}; \boldsymbol{v})}{\log\frac{|\mathcal{V}|}{N_{max}(t)}}, \tag{16}$$

*where $t' = t + c_1\sqrt{g_2(\delta, I(\boldsymbol{w}; \boldsymbol{v})) \cdot I(\boldsymbol{s}; \boldsymbol{v})} + c_2$. $g_1(\delta, I(\boldsymbol{w}; \boldsymbol{v})) = \delta(\frac{I(\boldsymbol{w};\boldsymbol{v})+1}{H(\boldsymbol{w})})$ $g_2(\delta, I(\boldsymbol{w}; \boldsymbol{v})) = (1-\delta)\frac{H(\boldsymbol{w}|\boldsymbol{v})-1}{H(\boldsymbol{w})}$*

## C    EXTRA LAYERS FOR FILTERING

### C.1    PROOF OF THEOREM 4.1

**Fact C.1** (Set disjointness communication lower bound (Yao, 1979)). *Suppose Alice and Bob are given inputs $a, b \in \{0, 1\}^n$, respectively, with the goal of jointly computing $\mathrm{DISJ}(a, b) = \max_i a_i b_i$ by alternately sending a single bit message to the other party over a sequence of communication rounds. Any deterministic protocol for computing $\mathrm{DISJ}(a, b)$ requires at least $n$ rounds of communication.*

$$r_i = \begin{cases} 0 & \text{if } \exists\, a, b \text{ s.t. } g(x_i, x_a, x_b) = 0 \\ 1 & \text{else} \end{cases}$$

In normal cases, judging the value of $r_i$ requires calculating $g(x_i, x_a, x_b)$ for all $a \in [0, n_d)$ and $b \in [n_d, n_d + n_q)$. Here we simplify the question, and we consider the situation where $g(x_i, x_a, x_b) = 0$ only if $b = a + n_d$ and $n_q = n_d$. Apparently, this is a special case of the original problem, and if one layer of self-attention fail to solve this, it is impossible for it to solve the original problem.

If we assume that the input is like,

$$\boldsymbol{x}_i \in \begin{cases} \{\boldsymbol{x}_i\} & \text{if } i = 0, \\ \{0, \boldsymbol{x}_a\} & \text{if } i \in \{1, \ldots, n_d - 1\}, \\ \{0, \boldsymbol{x}_b\} & \text{if } i \in \{n_d, \ldots, 2 \cdot n_d - 1\}. \end{cases} \tag{17}$$

Given input $(a, b) \in \{0, 1\}^{n_d} \times \{0, 1\}^{n_d}$, let $x_i = x_a$ if and only if $a_i = 1$ and let $x_i = x_b$ if and only if $b_{i-n_d} = 1$. In this way $r_i = 0$ if and only if $\mathrm{DISJ}(a, b) = 1$.

For simplicity, we use $n = n_d$. Now consider a more complex situation where Alice and Bob each hold a matrix $\boldsymbol{A} \in \mathbb{R}^{n \times d}$, $\boldsymbol{B} \in \mathbb{R}^{n \times d}$. each row of the matrix contains $\boldsymbol{w}$, so $d = H(\boldsymbol{w})$. and we assume that $\boldsymbol{x} = [\boldsymbol{s}, \boldsymbol{w}]$.

Then

$$\boldsymbol{x}_i = \begin{cases} \boldsymbol{x}_i & \text{if } i = 0, \\ [\boldsymbol{s}, \boldsymbol{a}_i] & \text{if } i \in \{1, \ldots, n_d - 1\}, \\ [\boldsymbol{s}, \boldsymbol{b}_i] & \text{if } i \in \{n_d, \ldots, 2 \cdot n_d - 1\}. \end{cases} \tag{18}$$

Let $\mathrm{DISJ1}(\boldsymbol{A}, \boldsymbol{B}) = \max_i(g'(\boldsymbol{x}, \boldsymbol{a}_i, \boldsymbol{b}_i))$, where the function $g'$ acts similar with $g$, but it takes $\boldsymbol{a}, \boldsymbol{b}$ as input, and $\boldsymbol{x}$ is a fixed constant as $\boldsymbol{x}_i$. So the calculation of $DISJ1$ requires $n \times H(\boldsymbol{w})$ bits of communication.

Also similar to the setting of $x$, $r_i = 1$ if and only if $\mathrm{DISJ1}(\boldsymbol{A}, \boldsymbol{B}) = 1$.

Then, this is the same with the 3Match problem, following the proof of Theorem 7 in Sanford et al. (2024), and with the following form of transformer $f(\boldsymbol{X})$, $2pH \log \log n + mpH \log \log n \approx mpH \log \log n$ bits are communicated.

$$f(\boldsymbol{X}) = \phi(f_h(\boldsymbol{X})),$$

where $\phi$ stands for the feed forward layers.

$$f_h(\boldsymbol{X}) = \frac{\sum_{i=1}^{N} \exp\left((\boldsymbol{W}_q x_1)^T \boldsymbol{W}_k x_i\right) \boldsymbol{W}_v x_i}{\sum_{i=1}^{N} \exp\left((\boldsymbol{W}_q x_1)^T \boldsymbol{W}_k x_i\right)}$$

Therefore, only we require $mpH \log \log n \geq nH(\boldsymbol{w}) \rightarrow mph \geq nH(\boldsymbol{w})/\log \log n$.

**Theorem C.2.** *For input documents of length $n$, if $mpH \leq \Omega(nH(\boldsymbol{w})/\log \log n)$, then there is no one layer transformer $\mathcal{M}$ with embedding size $m$, precision $p$ and $H$ heads satisfying $\mathcal{M}(X) = r$.*

Also, as shown in Sanford et al. (2024), multiple layers of multi-headed attention are subject to the same impossibility

**Conjecture C.3.** *Every multi-layer transformer that computes Match3 must have width, depth, embedding dimension, or bit complexity at least $N^{\Omega(1)}$.*

However, one layer of transformer can effectively solve Match2 problem, for RAG, we can simply fix one or some token to represent the information of a document, in this way $g(x_i, x_a, x_b)$ becomes $g(x_i, x_0, x_b)$, if we use the first transformer layer to aggregrate information of $x_i$ and $x_0$, then use the second layer to gather $x_b$ and judging the relevance by MLP layers, then the problem can be solved.

therefore, $g(x_i, x_a, x_b) = g(x_i, x_a, x_b) = g'(x_i', x_b')$. Then the problem becomes a pair-wise problem, and transformer can easily solve it.

**Proposition C.4.** *Using transformer to solve* $r_i = \begin{cases} 0 & \text{if } \exists \, a, b \text{ s.t. } g'(x_i', x_b') = 0 \\ 1 & \text{else} \end{cases}$ *requires* $mpHD \geq n_d H(\boldsymbol{w})$.

This mainly dues to the self regression of LLM, token $x_i$ can not gather information of token $x_j$ if $j > i$, which means that all the judgement of is done during the calculation of query rather than the document tokens because we put the documents ahead of the query. However, if we put the query first, then $g'(x_i, x_b)$ can be effectively calculated in the document token.

# D EXPERIMENTS

We conduct experiments using the multi-hop dataset (Ho et al., 2020; Yang et al., 2018; Trivedi et al., 2022). We utilize BM25 to extract documents, treating those that do not contain the middle reasoning results or the answer as distracting documents (Cuconasu et al., 2024). Due to the limited computational resource, we only sample 1000 queries to test the performance.

we use different task instruction when query is ahead or after the documents.

- query ahead document: *"You are given a question and you MUST respond with a short answer (max 5 tokens) based on the provided documents. If none of the documents contain the answer and you do not know the answer, please respond with NO-RES."*

- query after document: *You are given a question and you MUST respond with a short answer (max 5 tokens) based on the provided documents. If none of the documents contain the answer and you do not know the answer, please respond with NO-RES. The question will be presented both before and after the documents.*

Also, in the query after document situation, we actually also put the query after the documents to make LLM remember what the question is.

We conduct experiments on a single Nvidia RTX 3090 24G, the rank for LoRA finetuning is $64$, and we use 1,8,15 virtual tokens for prompt tuning and find the best performing one. We finetuning the learning rate with in $\{5e-7, 1e-6, 3e-6, 5e-6, 8e-6\}$.

We use three models to show the performance.

- meta-llama/Meta-Llama-3.1-8B-Instruct
- lmsys/vicuna-7b-v1.5
- mistralai/Mistral-7B-Instruct-v0.3

Due to limited computational resources, we sample 3,000 queries for training, and we use the code of Jeong et al. (2024) to extract those gold and distracting documents.

### D.1 EXPERIMENTS ON GOLD ONLY DATASETS

We finetune the model on datasets where gold documents and 3 more distracting documents are included in the context, and we want to show that how will the model perform on datasets with only gold documents, to show that LoRA finetuned model may lose some of its original functionality and our model DPrompt Tuning can relieve its negative impact by making the filtering process simpler.

Table 3: Results on pure gold datasets

|  | vanilla | Lora | Prompt | Lora+Prompt | Dprompt |
|---|---|---|---|---|---|
| 2wiki | 60.3 | 59.7 | 59.2 | 58.5 | 59.7 |
| hotpotqa | 62.7 | 55.4 | 57.2 | 54.8 | 60.3 |
| musique | 28.7 | 24.7 | 25.3 | 24.3 | 27.3 |

Those finetuned models are tested on pure gold where only gold documents are included. And we can observe that those finetuned models performs worse on pure gold setting because they pay more attention to filter noise and DPrompt Tuning can relieve the negative impact because it makes the filtering process easier, so the performance could be better.

## E  RELATED WORK

**RAG in LLM.** Many recent works have explored better RAG strategies. Shi et al. (2023b) treat the model as a black box and design a retriever to enhance performance. However, Shi et al. (2023a) found that distracting documents in the retrieved set significantly weaken performance, leading some to address this by fine-tuning the LLM (Yoran et al., 2023; Zhang et al., 2024a). Additionally, researchers have identified that noise in the prompt can negatively impact performance, prompting efforts to eliminate this noise and compress prompts (Jiang et al., 2023b;a; Pan et al., 2024).

**Filtering the noise.** Yoran et al. (2023) tries to finetune the LLM to better filter out distracting documents and achieve better performance, while RAFT (Zhang et al., 2024a) uses different proportion of distracting documents and finetune the model to better understand the question. Self-RAG(Asai et al., 2023), use special tokens to represent that should the LLM use document information and achieve great performance. Xu et al. (2024) first highlighted the duality of RAG, encompassing both benefits and detriments. Understanding how these factors influence the reasoning capabilities of LLMs remains an open question. Wang et al. (2023) also try to filter out irrelevant documents by information-theoretic approaches while Islam et al. (2024) decompose the LLM to a MoE model and use one expert to conduct filtering. Also some methods (Glass et al., 2022; Jiang et al., 2023b; Dong et al., 2024; Zhang et al., 2024b) try to recognize the relevance of documents and place the more relevance one in the first one or the last due to the Lost in the Middle phenomenon (Liu et al., 2024).

**Expressive Power of LLM.** Merrill & Sabharwal (2023) proves that the expressive power of LLM is constrained to $TC^0$, but Feng et al. (2024); Li et al. (2024) shows that by Chain of Thought prompting, we can infinitely stack transformer layers to solve any DP problems, Zeng & Lee (2023) further prove the rank needed to adapt the model for other takss by LoRA, and Petrov et al. (2024)

shows that prompt tuning can be a universal approximator with enough prompt length, while prompt tuning fail to change the attention pattern and can only bias the output (Petrov et al., 2023). RAG is also one kind of prompting technique, and it also try to involve more information when conducting inference like CoT. But how does RAG helps to improve the expressive power of LLM has not been studied before.

