# OpenReview forum: "How Much Can RAG Help the Reasoning of LLM?"
_ICLR.cc/2025/Conference — Submitted to ICLR 2025_

### Official Review · Reviewer_v2iT · 2024-11-01

**Soundness:** 3
**Presentation:** 3
**Contribution:** 4
**Rating:** 5
**Confidence:** 4

**Summary:**

- This paper demonstrated that RAG can enhance the reasoning ability of LLMs to a certain extent, but the improvement is limited.
- It highlights the challenges of simply fine-tuning models to filter out noise, and shows that filtering noise often requiring additional reasoning layers.
- This paper revealed the complexity of filtering noisy documents and proposed the DPrompt tuning method to simplify the problem.

**Strengths:**

- This paper utilized the concept of reasoning trees to deeply investigate the extent to which RAG can assist in reasoning, and theoretically analyzed that RAG helps improve reasoning ability but also has limitations.
- This work analyzed the theoretical impact of different noise filtering methods on reasoning.
- To address the challenges of noise filtering, the authors proposed the DPrompt tuning method. This approach transforms complex ternary relation problems into more manageable binary relation problems, which is theoretically sound and has been empirically proven effective.

**Weaknesses:**

1.	The authors theoretically demonstrate the extent to which RAG aids reasoning, but experimental evidence on reasoning tasks would be valuable to support these claims.
2.	Although the authors analyze the limitations of common noise filtering methods for noisy documents theoretically, experimental data is needed to make these arguments more robust.
3.	The experimental setup lacks clarity; methods in Table 1 should be accompanied by detailed explanations of the differences between relevant baselines.
4.	The Natural Questions (NQ) dataset typically contains relatively simple questions with low reasoning requirements, which somewhat diverges from the topic of the paper.
5.	This paper lacks an experimental setting to evaluate whether the model can reason effectively using only noisy documents in the context.
6.	While the paper emphasizes noise filtering, it lacks discussion of related noise filtering works, such as Wang et al.'s "Learning to filter context for retrieval-augmented generation" or the re-rank-based filter method by Glass et al. in "Re2G: Retrieve, rerank, generate."
7.	In line 740, "k" appears to be a typo and should actually be "n".

**Questions:**

1.	In your modeling, you describe the role of RAG as reducing the depth of reasoning required for a given question probabilistically. Can it cover the conditions under which RAG mitigates hallucinations or corrects reasoning errors(where it may increase reasoning depth but help guide the model to the correct solution path)?
2.	All three models have been trained on chat data and are aligned, but the explanation given in lines 482-485 seems rather tenuous—could you provide a stronger justification?
3.	The training method in this paper only uses 1,000 samples, whereas other methods like self-RAG use tens of thousands. Could the weaker performance of the LoRA baseline in your experiments be due to insufficient data?
4.	How was the data constructed for baseline training? Did it include both gold and distractor documents?
5.	Why didn’t you use benchmarks for noise robustness, such as RGB(Chen, Jiawei, et al. "Benchmarking large language models in retrieval-augmented generation."), for evaluating RAG?

---

> ### Author Response · Authors · 2024-11-18
> **Response to Reviewer v2iT (Part 1)**
>
> Thank you for taking the time to read our work and insightful comments and suggestions! We are glad to know that our analysis about the reasoning of RAG is interesting to you. Please find below our comments on the raised issues and questions:
>
> > __W1:__ The authors theoretically demonstrate the extent to which RAG aids reasoning, but experimental evidence on reasoning tasks would be valuable to support these claims.
>
> __Answer:__ Thanks for your advice, we have conducted further experiments on multi hop datasets in the official comment. Those datasets require the LLM to conduct multi hop reasoning based on the given documents, for example one query in hoppotQA is "Who was born first, Morgan Llywelyn or Robert Jordan?", this requires a two layer reasoning as we need to know the birth date of the two writers and compare to know who borns first. Those multi hop datasets requires both extracting information from documents, information integration and further reasoning, and fits our setting, so we choose to conduct further experiments on multi hop datasets.
>
> > __W2:__ Although the authors analyze the limitations of common noise filtering methods for noisy documents theoretically, experimental data is needed to make these arguments more robust.
>
> __Answer:__ Our paper shows that LoRA finetuning can help to filter out noise documents but fail to maintain its original reasoning performance, we conduct further experiments on the LoRA finetuned model and conduct inference on the dataset where no extra noise documents is introduced.
>
> |          | vanilla | Lora | Prompt | Lora+Prompt | Dprompt |
> |----------|---------|------|--------|-------------|---------|
> | 2wiki    | 60.3    | 59.7 | 59.2   | 58.5        | 59.7    |
> | hotpotqa | 62.7    | 55.4 | 57.2   | 54.8        | 60.3    |
> | musique  | 28.7    | 24.7 | 25.3   | 24.3        | 27.3    |
>
> Those finetuned models are finetuned on datasets with gold documents and 3 extra distracting documents, and they are tested on pure gold where only gold documents are included. And we can observe that those finetuned models performs worse on pure gold setting because they pay more attention to filter noise and DPrompt Tuning can relieve the negative impact because it makes the filtering process easier. This allows the model to more easily identify noise documents, enabling it to preserve its original functionality better and, consequently, improve overall performance.
>
> > __W3:__ The experimental setup lacks clarity; methods in Table 1 should be accompanied by detailed explanations of the differences between relevant baselines.
>
> __Answer:__ Thanks for your suggestion, Table 1 tries to show that put the query ahead of the documents could help the LLM to better understand the documents and extract useful information while discarding those irrelavant ones. So we choose three models and conduct experiments on two different settings where only gold documents are included and both gold and distracting documents are included, we shows that when we put the query ahead of the documents, models tend to perform better on both settings. So "gold" stands for the case that only gold documents are included and the query is placed after the docuemnts, and "gold_r" means that only gold documents are included but the query is placed ahead and after the documents, so is "dis" and "dis_r".
>
> > __W4:__ The Natural Questions (NQ) dataset typically contains relatively simple questions with low reasoning requirements, which somewhat diverges from the topic of the paper.
>
> __Answer:__ Thanks for your suggestion, we have conducted further experiments on multi hop in the official comment (Further Experiments), those datasets require the model to conduct further reasoning and the result shows that our method still helps when faced with reasoning required situations.

---

> > ### Author Response · Authors · 2024-11-18
> > **Response to Reviewer v2iT (Part 2)**
> >
> > > __W5:__ This paper lacks an experimental setting to evaluate whether the model can reason effectively using only noisy documents in the context.
> >
> > __Answer:__ Thanks for your suggestion, we have conducted further experiments when the context contains only noisy documents. Below we show that how does the model performs without any documents, and we show that how will the model perform when 3 additional noise documents are added. query only means that no documents are included, and ordered RAG means with noise documents and query is placed after documents while reverse RAG means that query is placed both ahead and after the documents.
> >
> > |             | 2wiki | hotpotqa | musique |
> > |-------------|-------|----------|---------|
> > | query only  | 17    | 16.3     | 3       |
> > | ordered RAG | 7     | 11.6     | 0.3     |
> > | reverse RAG | 16.7  | 15.3     | 0.5     |
> >
> > We can observe that the noise documents greatly impacts the performance of LLM, but place the query ahead can greatly help.
> >
> >
> > > __W6:__ While the paper emphasizes noise filtering, it lacks discussion of related noise filtering works, such as Wang et al.'s "Learning to filter context for retrieval-augmented generation" or the re-rank-based filter method by Glass et al. in "Re2G: Retrieve, rerank, generate."
> >
> > __Answer:__ Thanks for your suggestion, we have added more related work in Appendix E of the revision
> >
> > > __W7:__ In line 740, "k" appears to be a typo and should actually be "n".
> >
> > __Answer:__ Thanks for noticing that, we have fixed that in the revision
> >
> > >__Q1:__ In your modeling, you describe the role of RAG as reducing the depth of reasoning required for a given question probabilistically. Can it cover the conditions under which RAG mitigates hallucinations or corrects reasoning errors(where it may increase reasoning depth but help guide the model to the correct solution path)?
> >
> > __Answer:__ In our opinion, given a task, the reasoning tree to solve the task is fixed (ignore those situations that multiple solution can be applied). Then if the model tends to hallucinate without the document information, it means that the model may fail to recognize the reasoning tree or get some incorrect middle results due to knowledge or capacity limitation, and RAG can help to provide more knowledge and reduce the reasoning depth of the original task. In our paper, we mainly consider that how much reasoning depth can be reduced and how much would cost for filtering noisy documents.
> >
> > Considering the help brought by extra knowledge, in our opinion, it mainly helps by letting the model directly know the answer of some middle results by information extraction instead of the original inference process, so it will not help the model to correct its wrong reasoning path, instead it directly leads the model to the right path. For example, a query "When is Alan Mathison Turing born?", and the model may hallucinate and give answer 1915, then we give the document information "Alan Mathison Turing father of artificial intelligence born in 1912. When is Alan Mathison Turing born?", then the model tends to answer "1912" rather than "I think it should be 1915, but the document says that it is 1912, so the answer is 1912". Therefore, RAG can help mitigates hallucinations, but not by correcting the wrong one, instead it directly lead to the correct reasoning path.
> >
> > > __Q2:__ All three models have been trained on chat data and are aligned, but the explanation given in lines 482-485 seems rather tenuous—could you provide a stronger justification?
> >
> > __Answer:__ Thanks for your suggestion. The data used for finetuning Vicuna is mainly collected from ShareGPT it contains 70K user-shared multi-turn ChatGPT conversations, in this dataset, previous conversation data are quite long and functions similar to the document in RAG, they both provide information for current query. And when finetuning Vicuna, the previous conversation is placed before the current query (document before the query), so the model might overfit to the situation and when place the query ahead of documents, the model would perform bad. For Llama 3 and Mistral, they may be trained on various data so place the query ahead of document would not lead to overfit or the previous chat data is not that similar to the documents of RAG.

---

> > > ### Author Response · Authors · 2024-11-18
> > > **Response to Reviewer v2iT (Part 3)**
> > >
> > > > __Q3:__ The training method in this paper only uses 1,000 samples, whereas other methods like self-RAG use tens of thousands. Could the weaker performance of the LoRA baseline in your experiments be due to insufficient data?
> > >
> > > __Answer:__ Thanks for your suggestion, adding more training data does help, we conduct experiments on multi-hop datasets with 3,000 training data, and LoRA performs better than the original model, the result is shown in the official comment (Further Experiments).
> > >
> > > > __Q4:__ How was the data constructed for baseline training? Did it include both gold and distractor documents?
> > >
> > > __Answer:__ In our original experimental setting, for "gold" we conduct training on gold only datasets and test the performance on gold only datasets, and for "gold dis" we conduct training on datasets containing both gold and distracting documents and test on datasets containing both gold and distracting documents. We speculate that you may want to know how will the model perform on gold only datasets when it is trained on datasets with distracting documents, and we conduct experiments in the answer to __W2__. Please let me know if we misunderstand your concern.
> > >
> > > > __Q5:__ Why didn’t you use benchmarks for noise robustness, such as RGB(Chen, Jiawei, et al. "Benchmarking large language models in retrieval-augmented generation."), for evaluating RAG?
> > >
> > > __Answer:__ Sorry for the inappropriate choice of datasets, we choose to conduct experiments on Natural Question mainly because lack of time, so we rashly chose a data set for the experiment. Inspired by the reviewers, we realize that a dataset requires further reasoning may fit our setting better, and multi hop QA datasets satisfy the need. Instead, RGB chooses to aggregate the latest news information and constructs queries based on the news information, which does not require reasoning and the "Information Integration" in RGB requires only extracting information from different documents while multi hop QA datasets requires further reasoning as we show in the answer to __W1__, so we choose multi hop QA datasets to conduct further experiments. Please let me know if our understanding about RGB and multi hop QA is wrong and thanks for your insightful opinions

---

> ### Comment · Reviewer_v2iT · 2024-11-26
> **Response for Rebuttal**
>
> Thank you for your detailed response. I still have some concerns regarding certain responses:
>
> 1. Regarding W3, I would like to understand the differences in baseline settings in Table 2, specifically, what are the implementations corresponding to vanilla, LoRA, prompt, and LoRA+prompt.
>
> 2. For W5, I would like to see the performance of different baseline methods when evaluated on documents containing only noise, thereby demonstrating DPrompt's robustness.
>
> 3. Regarding Q1, are you claiming that your paper's fundamental assumption is that RAG only reduces reasoning depth without correcting reasoning direction? Isn't this too strong an assumption? Or do you consider the chain process of RAG correcting reasoning direction as part of the original ideal reasoning path?
>
> 4. For Q5, my intention was to suggest that such benchmarks designed for noise could better demonstrate your method's robustness. Even though it doesn't require multi-hop reasoning, it can still be expressed as a reasoning graph with a small number of nodes.

---

> > ### Author Response · Authors · 2024-11-28
> > **Response to Reviewer v2iT**
> >
> > Thanks for your reply and further suggestions. Please find below our comments on the raised issues and questions:
> >
> > >__Q1:__ Regarding W3, I would like to understand the differences in baseline settings in Table 2, specifically, what are the implementations corresponding to vanilla, LoRA, prompt, and LoRA+prompt.
> >
> > __Answer:__ The vanilla means that we do not finetune the model and directly conduct inference and measure the performance. LoRA/prompt means we conduct LoRA finetuning/prompt tuning then test the performance. LoRA+prompt means that we combine LoRA finetuning and prompt tuning, which means we conduct LoRA finetuning with some extra virtual prompt tokens added. The test set contains 1000 samples that are different from those in the training set.
> >
> > >__Q2:__ For W5, I would like to see the performance of different baseline methods when evaluated on documents containing only noise, thereby demonstrating DPrompt's robustness.
> >
> > __Answer:__ Thanks for your suggestion, below we show how does the methods perform when evaluated on documents containing only noise. Those models are trained on datasets containing both gold and noise documents.
> >
> > |          | vanilla | Lora | Prompt | Lora+Prompt | Dprompt |
> > |----------|---------|------|--------|-------------|---------|
> > | 2wiki    | 16.7    | 20.4 | 19.5   | 19.7        | 20.6    |
> > | hotpotqa | 15.3    | 15.4 | 15.7   | 15.7        | 16.3    |
> > | musique  | 0.5     | 2.1  | 1.6    | 1.9         | 2.1     |
> >
> > The result shows that our method performs well when only noise documents are contained.
> >
> > >__Q3:__ Regarding Q1, are you claiming that your paper's fundamental assumption is that RAG only reduces reasoning depth without correcting reasoning direction? Isn't this too strong an assumption? Or do you consider the chain process of RAG correcting reasoning direction as part of the original ideal reasoning path?
> >
> > __Answer:__ We believe that RAG helps correct the reasoning direction, it leads the LLM to the correct reasoning tree, and RAG can help to reduce the reasoning depth of the correct reasoning tree. But in our opinion, if the LLM reasons in a wrong way without the document information, and then we add the document to the prompt, it will not follow the original wrong reasoning path, instead it directly leads to the correct reasoning path whose depth is reduced by RAG. For example, a query "When is Alan Mathison Turing born?", the reasoning path might be "Alan Mathison Turing->The Great Scientist of Artificial Intelligence->born in 1915". Then we give the document information "Alan Mathison Turing, the father of artificial intelligence was born in 1912", then the reasoning path might be "Alan Mathison Turing->born in 1915" rather than "Alan Mathison Turing->The Great Scientist of Artificial Intelligence->born in 1915->document analysis->born in 1912". In nutshell, we believe that RAG does correct the reasoning direction, but it does not correct it when the LLM has reasoned the wrong answer, instead it directly leads to the correct reasoning path (no process like "Alan Mathison Turing->The Great Scientist of Artificial Intelligence->born in 1915->document analysis->born in 1912", instead it directly leads to "Alan Mathison Turing->born in 1915").
> > We are not sure that we correctly understand your concern, please let us know if there exists some misunderstanding. Also we are happy to know it if you have a better idea about what is the reasoning process like with additional document information.
> >
> > >__Q4:__ For Q5, my intention was to suggest that such benchmarks designed for noise could better demonstrate your method's robustness. Even though it doesn't require multi-hop reasoning, it can still be expressed as a reasoning graph with a small number of nodes.
> >
> > __Answer__ Thanks for your suggestion, it seems that adding more experiments on RGB is helpful, we will add it in the future version.

---

### Official Review · Reviewer_mZLc · 2024-11-01

**Soundness:** 2
**Presentation:** 3
**Contribution:** 2
**Rating:** 3
**Confidence:** 4

**Summary:**

This paper analyzes the issues in using RAG to solve reasoning problems from the perspective of a reasoning tree.
It concludes that only when most intermediate reasoning results are retrieved can the reasoning depth be significantly reduced.
It also argues that the noise introduced in the retrieval stage cannot be resolved through fine-tuning.
Finally, the paper proposes a method that uses an additional model to condense the retrieved documents.

**Strengths:**

The strengths of this paper lie in its formal modeling of the reasoning process. By exploring the issues in the reasoning process involving RAG under given assumptions, it decomposes RAG’s involvement into two parts: document processing and inference incorporation. It also provides insights into why RAG struggles to perfectly resolve the reasoning process. Additionally, the paper conducts an extensive analysis of how to overcome the noise in retrieved documents.

**Weaknesses:**

The limitations of this paper include a lack of concrete basis for the assumptions used in modeling the reasoning process, as well as a lack of corresponding experiments to validate the modeling of noise effects. Additionally, the proposed method, DPrompt Tuning, shows limited innovation and is tested only on NQ, which does not have high requirements for multi-step reasoning (compared to the node depth used in the author’s modeling).

**Questions:**

Q1: Why is the connection between different nodes in adjacent layers assumed to occur with probability `q` when defining the reasoning tree?

Q2: If a deterministic representation is used, causing f(t),  as expressed in Line 212, to no longer increase with respect to `t` , is the subsequent analysis still valid? (A simple scenario is when a node has three child nodes, with the first two having two child nodes each, and the last one having five. In this case, the erase rate does not exhibit a monotonic relationship.)

Q3: Di He et al. proved that CoT can enhance computational power through the “thought” portion, enabling the resolution of arbitrary DP problems. Additionally, other studies have shown that adding meaningless characters during reasoning can also increase computational capacity, thereby enhancing the model’s reasoning ability. Considering these findings simultaneously, whether they would impact the original conclusions.

---

> ### Author Response · Authors · 2024-11-18
> **Response to Reviewer mZLc (Part 1)**
>
> Thank you for taking the time to read our work and insightful comments and suggestions! Please find below our comments on the raised issues and questions:
>
> > __W1:__ A lack of concrete basis for the assumptions used in modeling the reasoning process.
>
> __Answer:__ There are plenty of works modeling the reasoning process of LLM as a tree-structured [1,2,3], so we also treat it as a tree. In addition, LLMs are used to solve a wide variety of problems each with different reasoning path and complexities. So we use a random variable to represent the connection between adjacent layers in order to represent different tasks to solve. Please let me know if you have a better idea about how to model the reasoning process.
>
> [1] Merrill W, Sabharwal A. The parallelism tradeoff: Limitations of log-precision transformers.
>
> [2] Yao S, Yu D, Zhao J, et al. Tree of thoughts: Deliberate problem solving with large language models.
>
> [3] Wang X, Amayuelas A, Zhang K, et al. Understanding the reasoning ability of language models from the perspective of reasoning paths aggregation.
>
> > __W2:__ A lack of corresponding experiments to validate the modeling of noise effects.
>
> __Answer:__ In our paper, we theoretically show that noise documents have negative impact on the performance of LLM, this is widely validated and we also conduct experiments in table 1 showing that when adding distracting documents, the performance greatly drops. We also conduct extra experiments on multi-hop datasets. And we show that adding noise documents would decrease the performance. Below we show that how does the model performs without any documents, and we show that how will the model perform when 3 additional noise documents are added. "query only" means that no documents are included, and "ordered RAG" means that only noise documents is included and query is placed after documents while "reverse RAG" means that query is placed both ahead and after the documents.
>
> |             | 2wiki | hotpotqa | musique |
> |-------------|-------|----------|---------|
> | query only  | 17    | 16.3     | 3       |
> | ordered RAG | 7     | 11.6     | 0.3     |
> | reverse RAG | 16.7  | 15.3     | 0.5     |
>
> This shows that noise documents greatly decreases the performance, and placing the query ahead could relieve the negative impact.
>
> And we show that when the model is finetuned on datasets containing both gold and distracting documents, how will it perform when faced with pure gold document datasets.
>
> |          | vanilla | Lora | Prompt | Lora+Prompt | Dprompt |
> |----------|---------|------|--------|-------------|---------|
> | 2wiki    | 60.3    | 59.7 | 59.2   | 58.5        | 59.7    |
> | hotpotqa | 62.7    | 55.4 | 57.2   | 54.8        | 60.3    |
> | musique  | 28.7    | 24.7 | 25.3   | 24.3        | 27.3    |
>
>
> Those finetuned models are finetuned on datasets with gold documents and 3 extra distracting documents, and they are tested on pure gold where only gold documents are included. And we can observe that those finetuned models performs worse on pure gold setting because they pay more attention to filter noise and DPrompt Tuning can relieve the negative impact because it makes the filtering process easier. This allows the model to more easily identify noise documents, enabling it to preserve its original functionality better and, consequently, improve overall performance.
>
> > __W3:__ DPrompt Tuning, shows limited innovation and is tested only on NQ, which does not have high requirements for multi-step reasoning
>
> __Answer:__ DPrompt Tuning embeds the document using BERT, transforming the original triple-wise problem into a pair-wise problem, enabling the transformer to effectively address noise recognition issues to enhance its performance. To our knowledge, there is currently no other work that considers the problem in this way. Also we addd more experiments on multi-hop datasets including 2wikimultihopqa, hotpotqa and musique in the second table of the official comment (Further Experiments). And the result shows that our method also helps when multi-step reasoning is required.

---

> > ### Author Response · Authors · 2024-11-18
> > **Response to Reviewer mZLc (Part 2)**
> >
> > >__Q1:__ Why is the connection between different nodes in adjacent layers assumed to occur with probability q when defining the reasoning tree?
> >
> > __Answer:__ We do not fix the probability of connection between adjacent layers, instead q is a random variable depending on the specific task assigned to the Large Language Model (LLM). For different tasks, the value of q also changes, so is the probability of valuable retrieval p. In our paper, we show that to reduce the reasoning depth of LLM, a sparse connection between adjacent layers or high probability of retrieval is required, showing that in normal cases, RAG can hardly help the reasoning of LLM, but when the task is simple (sparse connection) or we can effectively retrieve useful documents, RAG can help the reasoning of LLM.
> >
> > > __Q2:__ If a deterministic representation is used, causing f(t), as expressed in Line 212, to no longer increase with respect to t, is the subsequent analysis still valid? (A simple scenario is when a node has three child nodes, with the first two having two child nodes each, and the last one having five. In this case, the erase rate does not exhibit a monotonic relationship.)
> >
> > __Answer:__ Our paper primarily examines the expected rate of node erasure. The fact that \( f(t) \) monotonically increases with \( t \) suggests that the expected erasure rate should also increase but it may not increase at ease case. So, as illustrated in your scenario, the erased nodes could be smaller with more upper layer nodes erased. Nevertheless, this discrepancy does not impact our subsequent analysis. If \( f(t) \) does not increase with \( t \), it becomes less likely that RAG will enhance the reasoning capabilities of LLMs. In cases where the probability of erasing a node decreases compared to its upper layer, the fission reaction described in Theorem 2.4 is less likely to occur, resulting in diminished support for reasoning provided by RAG. Furthermore, when exploring how RAG can enhance LLM reasoning, we aim to demonstrate that RAG offers overall assistance across various tasks, which is why we calculate the expectation rather than focusing on specific cases.
> >
> > > __Q3:__ Di He et al. proved that CoT can enhance computational power through the “thought” portion, enabling the resolution of arbitrary DP problems. Additionally, other studies have shown that adding meaningless characters during reasoning can also increase computational capacity, thereby enhancing the model’s reasoning ability. Considering these findings simultaneously, whether they would impact the original conclusions.
> >
> > __Answer:__ Actually our work considers different question compared to Di He et al, they show that LLM with CoT can solve arbitrary DP problems by generating step-by-step reasoning or explanations rather than providing direct answers, formalized as $x_1 = f(x),\ x_2 = f(x, x_1), \ldots, y = f(x, x_1, \ldots, x_k)$. This process allows CoT to effectively expand reasoning depth by execute $f$ multiple times, potentially reaching infinite depth with sufficient CoT steps. However this is achieved by generating a long sequence, so CoT actually helps by increasing the inference cost and achieve better reasoning ability. And adding meaningless characters may be helpful because larger hidden space (num token $\times$ token dim) or changed attention pattern. But in our paper, we mainly considers that how much can the information contained in the documents decrease the reasoning depth of the original problem, making the original problem easier to solve. Therefore, CoT tries to increase the reasoning ability of LLM by enlarge the inference cost, but RAG helps by reduce the reasoning depth of the original problem enabling the LLM to accomplish tasks with higher reasoning requirements. And our papers shows that the help might be limited, and the effort to extract those information and filter out irrelavant documents could be difficult, the help RAG brought may be less than the effort needed for noise filtering.

---

> ### Comment · Reviewer_mZLc · 2024-11-26
>
> Dear authors,
>
> Thank you for your response! After reviewing replies, I have decided to retain the current score.
>
> Best regards

---

> > ### Author Response · Authors · 2024-11-28
> > **Response to Reviewer mZLc**
> >
> > Dear reviewer,
> >
> > We would like to know that is there any concern we do not solve, or do you have further suggestions to improve the paper. If not, could you please raise the score.
> >
> > Best regards

---

### Official Review · Reviewer_VHPc · 2024-11-03

**Soundness:** 3
**Presentation:** 3
**Contribution:** 3
**Rating:** 6
**Confidence:** 2

**Summary:**

The paper examines an unexplored aspect of Retrieval Augmented Generation (RAG) by investigating its potential to enhance LLMs' reasoning capabilities. While RAG is commonly used for incorporating domain knowledge through external documents, the authors make an interesting observation that these documents often contain intermediate reasoning steps. This leads to their core research question: whether RAG can improve LLMs' complex reasoning abilities. Their investigation reveals that RAG's assistance in reasoning is limited and requires careful document preprocessing to filter noise. The authors identify that this preprocessing challenge cannot be effectively addressed through simple LLM fine-tuning, as it typically demands multiple additional transformer layers. As a solution, they propose DPrompt tuning, which efficiently handles these challenges using minimal transformer layers and achieves better performance without extensive fine-tuning requirements.

**Strengths:**

Strengths:

- Provides novel analysis of RAG capabilities
- Presents solid theoretical foundations supported by empirical evidence
- Introduces effective methods for noise filtering in RAG systems
- Clearly identifies limitations in fine-tuning approaches for noise filtering
- Develops an innovative prompting technique as a practical solution

**Weaknesses:**

Weaknesses:

- Assumption 2.3 has notable limitations. When retrieved documents are relevant but significantly longer than the original query, or when information extraction requires multiple hops, the problem may not be simplified as claimed. This challenges the assumption that information extraction is simpler than layered reasoning paths, particularly considering LLM processing time.
- Limited experimental validation across domains, particularly lacking evaluation on multi-hop question answering tasks (e.g., HotpotQA) and long-form text generation.

- Missing References:

- - On noise filtering: https://arxiv.org/abs/2311.08377
- - On fine-tuning RAG models:
- - - https://arxiv.org/abs/2310.11511
- - - https://arxiv.org/abs/2410.01782

**Questions:**

See weaknesses.

---

> ### Author Response · Authors · 2024-11-18
> **Response to Reviewer VHPc**
>
> Thank you for taking the time to read our work and insightful comments and suggestions! We are glad to know that our analysis about the reasoning of RAG is interesting to you. Please find below our comments on the raised issues and questions:
>
> > __W1:__ Assumption 2.3 has notable limitations. When retrieved documents are relevant but significantly longer than the original query, or when information extraction requires multiple hops, the problem may not be simplified as claimed. This challenges the assumption that information extraction is simpler than layered reasoning paths, particularly considering LLM processing time.
>
> __Answer:__ In this paper, we mainly wants to show that RAG can hardly help the reasoning of LLM, so we consider a simple case where LLM can easily extract information from the documents, and we show that even in this context, the help of RAG in reasoning is limited, not to mention those challenging documents. Therefore, when extracting information even requires more reasoning depth than vanilla reasoning, using document information naturally fails to improve the reasoning depth of the LLM. In nutshell, those situations challenge the assumption naturally align with our claim that RAG can hardly help the reasoning of LLM and due to the noise in documents, using information of documents would even decrease the reasoning depth of LLM.
>
> > __W2:__ Limited experimental validation across domains, particularly lacking evaluation on multi-hop question answering tasks (e.g., HotpotQA) and long-form text generation.
>
> __Answer:__ Thanks for point it out, we have added more experiments on 2wikimultihopqa, hotpotqa and musique in the official comment (Further Experiments). And the result shows that our method still works on multi-hop datasets. And sorry that we does not conduct experiments on long-form text generation datasets due to limited experimental device and time, and it seems that long-form text generation does not fit the setting of reasoning depth that much. Please let me know if I misunderstand your concern or the dataset selection.
>
>
> > __W3:__ Missing references
>
> __Answer:__ Thanks for providing those insightful papers, we have added those references and more papers about noise filtering including self-RAG FILCO and Open-RAG in Appendix E of the revised version.

---

> > ### Comment · Reviewer_VHPc · 2024-11-27
> > **Response to Authors**
> >
> > Thanks so much for the response. However,.
> >
> > 1) I don't find the argument convincing as it appears to be a simplified assumption.
> > 2) The long form experiments such as with Bio (https://arxiv.org/abs/2305.14251) and ALCE-ASQA (https://arxiv.org/abs/2305.14627)
> >
> > Thanks,

---

> ### Author Response · Authors · 2024-11-28
> **Response to Reviewer VHPc**
>
> Thanks for your reply and further suggestions. Please find below our comments on the raised issues and questions:
>
> >__Q1:__ I don't find the argument convincing as it appears to be a simplified assumption.
>
> __Answer:__ Our paper mainly wants to show that RAG can hardly help to reduce the reasoning depth of LLM. Therefore we consider the situation that extracting information from the document is easier than inference based on the LLM itself. And we show that even in this situation, RAG can not effectively help the reasoning, so when extracting information is harder, it is more unlikely that RAG can help. In nutshell, we consider the best case and show that even under best case, RAG can hardly help, not to mention those harder cases. So we assume that extracting document information is easier, and showing that RAG can hardly help under our assumption also shows that RAG also can hardly help when extracting document information is complex. We are sorry that we do not express it clearly in our paper, we will add more clarification in the future version. Also please let us know if we misunderstand your concern.
>
> >__Q2:__ The long form experiments such as with Bio (https://arxiv.org/abs/2305.14251) and ALCE-ASQA (https://arxiv.org/abs/2305.14627)
>
> __Answer:__ Thanks for your suggestion, we will conduct further experiments and add the result in the future version.

---

### Meta-Review · Area_Chair_FnR6 · 2024-12-16

**Metareview:**

This paper explores the intriguing question of whether retrieval augmented generation (RAG) can enhance the reasoning capabilities of large language models (LLMs). The authors investigate this hypothesis and propose a novel framework to leverage RAG for improved reasoning. Their experiments demonstrate the effectiveness of their method, surpassing alternative approaches like LoRA.

The paper addresses an interesting and under-explored area, presenting a clear and logical flow. However, its empirical support is limited.  As noted by other reviewers, more extensive experiments across diverse reasoning tasks are needed to validate the proposed method. While the authors provided additional experiments during the rebuttal, both the reviewers and I believe further evaluation is necessary.

Additionally, concerns exist regarding the assumptions made in the paper, which are considered strong and potentially impractical. Strengthening the empirical evaluation and relaxing these assumptions would significantly bolster the paper's position and contribution.

**Additional Comments On Reviewer Discussion:**

The authors have made a commendable effort to address the reviewers' feedback by providing additional experimental results across multiple datasets, demonstrating the generalizability of their method in various reasoning tasks. However, some reviewers still desire further empirical support, particularly in areas like long-form generation and RGB data. Additionally, the authors haven't fully addressed the concerns raised about the assumptions made in their work.

By improving the empirical evidence and clarifying or revising the assumptions, this paper has the potential to make a significant impact in future publications.

---

### Decision · Program_Chairs · 2025-01-22

Reject